# Comparative Generalization Bounds for Deep Neural Networks

**Tomer Galanti**                                                                                          *galanti@mit.edu*
*Department of Brain and Cognitive Sciences*
*Massachusetts Institute of Technology*

**Liane Galanti**                                                                              *lianegalanti@mail.tau.ac.il*
*School of Computer Science*
*Tel Aviv University*

**Ido Ben-Shaul**                                                                           *ido.benshaul@gmail.com*
*Department of Applied Mathematics*
*Tel-Aviv University*
*eBay Research*

**Reviewed on OpenReview:** *https://openreview.net/forum?id=162TqkUNPO*

## Abstract

In this work, we investigate the generalization capabilities of deep neural networks. We introduce a novel measure of the effective depth of neural networks, defined as the first layer at which sample embeddings are separable using the nearest-class center classifier. Our empirical results demonstrate that, in standard classification settings, neural networks trained using Stochastic Gradient Descent (SGD) tend to have small effective depths. We also explore the relationship between effective depth, the complexity of the training dataset, and generalization. For instance, we find that the effective depth of a trained neural network increases as the proportion of random labels in the data rises. Finally, we derive a generalization bound by comparing the effective depth of a network with the minimal depth required to fit the same dataset with partially corrupted labels. This bound provides non-vacuous predictions of test performance and is found to be empirically independent of the actual depth of the network.

## 1 Introduction

Deep learning systems have steadily advanced the state of the art in a wide range of benchmarks, demonstrating impressive performance in tasks ranging from image classification (Taigman et al., 2014; Zhai et al., 2022), language processing (Devlin et al., 2019; Brown et al., 2020), open-ended environments (Silver et al., 2016; Arulkumaran et al., 2019), to coding (Chen et al., 2021).

Recent research suggests that deep neural networks are able to generalize well to new data because they have a large number of parameters relative to the number of training samples (Belkin et al., 2018; Belkin, 2021; Advani et al., 2020; Belkin et al., 2019). However, it has been shown that in these cases deep learning models can also precisely interpolate arbitrary training labels (Zhang et al., 2017), a phenomenon known as the "interpolation regime." Understanding how deep learning models learn through interpolation is an important step towards a more comprehensive theoretical understanding of their successes.

Traditional generalization bounds (Vapnik, 1998; Shalev-Shwartz & Ben-David, 2014; Mohri et al., 2018; Bartlett & Mendelson, 2003) are based on uniform convergence and are used to control the worst-case generalization gap (the difference between train and test errors) over a set of predictors that includes the outputs

| Dataset | MNIST | | | Fashion MNIST | | | CIFAR10 | | | CIFAR10 | | |
|---|---|---|---|---|---|---|---|---|---|---|---|---|
| Architecture | CONV-$L$-50 | | | CONV-$L$-100 | | | CONV-$L$-100 | | | CONVRES-$L$-50 | | |
| Depth ($L$) | 10 | 12 | 15 | 10 | 12 | 15 | 16 | 18 | 20 | 10 | 12 | 15 |
| Test error | 0.0075 | 0.0074 | 0.0074 | 0.0996 | 0.0996 | 0.0996 | 0.2659 | 0.2653 | 0.2648 | 0.2903 | 0.2862 | 0.2804 |
| Test error std. | 0.0006 | 0.0009 | 0.0008 | 0.0026 | 0.003 | 0.0024 | 0.0144 | 0.0057 | 0.0045 | 0.0049 | 0.0067 | 0.0044 |
| $p$ | 0.1 | 0.1 | 0.1 | 0.2 | 0.2 | 0.2 | 0.4 | 0.4 | 0.4 | 0.4 | 0.4 | 0.4 |
| **Our bound** | **0.1** | **0.1** | **0.1** | **0.2** | **0.2** | **0.2** | **0.66** | **0.66** | **0.53** | **0.4** | **0.4** | **0.4** |
| $L_{1,\infty}$ (Bartlett & Mendelson, 2003) | 8.911e+14 | 1.74e+17 | 2.13e+22 | 3.613e+17 | 9.145e+18 | 4.088e+22 | 1.076e+23 | 6.682e+28 | 2.758e+35 | - | - | - |
| $L_{3,1.5}$ (Neyshabur et al., 2015) | 5.462e+05 | 1.6e+06 | 1.308e+06 | 7.523e+07 | 6.997e+07 | 2.636e+08 | 4.633e+08 | 2.275e+09 | 5.061e+09 | - | - | - |
| Frobenius (Neyshabur et al., 2015) | 1.848e+06 | 8.194e+06 | 2.216e+07 | 2.486e+08 | 2.335e+08 | 1.585e+09 | 1.967e+09 | 1.442e+10 | 3.038e+11 | - | - | - |
| Spec $L_1$ (Bartlett et al., 2017) | 2.861e+05 | 6.412e+05 | 9.566e+05 | 4.706e+06 | 3.516e+06 | 3.176e+06 | 1.19e+07 | 1.449e+08 | 1.272e+10 | - | - | - |
| Spec Frob (Neyshabur et al., 2019) | 3.948e+03 | 1.1199e+04 | 1.538e+04 | 4.0229e+04 | 2.884e+04 | 2.543e+04 | 9.4833e+04 | 1.011e+06 | 1.033e+08 | - | - | - |

Table 1: **Comparing our bound with baseline bounds in the literature for networks of varying depths.** Our error bound is reported in the fourth row, and the baseline bounds are reported in the bottom rectangle. While the test error is universally bounded by 1, the baseline bounds are much larger than 1, and therefore, are meaningless. **In contrast, our bound achieves relatively tight estimations of the test error and unlike the baseline bounds, our bound is fairly unaffected by the network's depth.**

of a learning algorithm. However, the applicability of these bounds to certain interpolation learning regimes has been called into question by Nagarajan & Kolter (2019), who described theoretical scenarios where an interpolation learning algorithm generalizes well but a uniform convergence bound cannot detect this. Subsequent research by Bartlett & Long (2021); Zhou et al. (2020); Negrea et al. (2020); Yang et al. (2021) has also demonstrated the limitations of uniform convergence in various interpolation learning situations.

Several recent papers have obtained non-vacuous bounds in various learning settings. For instance, Dziugaite & Roy (2017); Zhou et al. (2019) adopted a PAC-Bayesian setting where the network incorporates random weights, while Hellström & Durisi (2022) considers random compressed versions of the neural network. While these advancements have significantly contributed to the knowledge of generalization in deep learning theory, they are far from contemporary practice, which generally focuses on deterministic networks obtained directly through stochastic gradient descent (SGD). In a different paper (Biggs & Guedj, 2022), the authors derived deterministic generalization bounds for neural networks based on PAC-Bayes and stability. However, this analysis is limited to 2-layer fully-connected networks.

**Contributions.** In this paper, we present a novel approach for measuring generalization in deep learning that does not rely on uniform convergence bounds. Instead, our bound suggests that a model will perform well at test time if its complexity is small compared to the complexity of a network required to fit the same dataset with partially random labels. In other words, even if a trained network has a complexity greater than the number of training samples, it may still be less complex than a model that fits partially random labels. As a result, in such cases, our bound may provide a non-trivial estimate of the test error. In contrast to previous non-vacuous bounds in the literature, our bound applies to a wide range of neural networks, including convolutional layers, batch normalization layers, and residual connections.

To formally describe our notion of complexity, we use the concept of nearest class-center (NCC) separability. This property states that the feature embeddings associated with training samples belonging to the same class can be separated using the nearest class-center decision rule. While earlier research (Papyan et al., 2020) found that NCC separability occurs at the penultimate layer of trained networks, more recent research (Ben-Shaul & Dekel, 2022) has discovered NCC separability in intermediate layers as well. In this work, we introduce the concept of "effective depth" in neural networks, which refers to the lowest layer at which the features are NCC separable (see Sec. 3.2).

We have made several key observations about effective depths. First, we have found that the effective depth of trained networks increases as the amount of random labels in the data increases. Second, when training deep networks, we have observed that they tend to converge to an effective depth $L_0$, regardless of their actual depth $L$. This means that the feature embeddings of layers above $L_0$ tend to be NCC separable. In addition, we have shown in Tab. 1 that our bound on generalization is empirical, non-vacuous, and independent of depth, unlike traditional bounds. In Section 3.3 we further discuss the limitations of modern norm-based generalization bounds (e.g., Neyshabur et al. (2015); Bartlett et al. (2017); Golowich et al. (2020); Neyshabur et al. (2018)), along with the key distinctions between these bounds and the proposed bound.

## 1.1 Additional Related Work

**Emergent properties.** There has been significant research on the geometrical properties of intermediate layers in deep neural networks, such as clustering and separability (Papyan, 2020; Tirer & Bruna, 2022; Galanti et al., 2022; Ben-Shaul & Dekel, 2022; Cohen et al., 2018; Alain & Bengio, 2017; Montavon et al., 2011; Papyan et al., 2017; Ben-Shaul & Dekel, 2021; Shwartz-Ziv & Tishby, 2017). While previous studies have analyzed these properties theoretically (Zhu et al., 2021; Xu et al., 2023; Lu & Steinerberger, 2020; Fang et al., 2021; Ergen & Pilanci, 2021), their specific role in deep learning and potential relationship with generalization are not yet fully understood. We focus on the question of whether these properties are good indicator of generalization. In contrast, previous research (Zhu et al., 2021) has shown that such properties may occur even when training a network with random labels, suggesting that they may not directly indicate generalization. In this paper, we argue that effective depth can be used to measure the complexity of fitting a dataset, and show how this idea can help us predict test performance.

## 2 Problem Setup

In this section, we explain the learning setting used in our theory and experiments. We focus on the task of training a model for standard multi-class classification. Specifically, we consider a distribution $P$ over samples $(x, y)$ where $x$ belongs to the instance space $\mathcal{X}$, and $y$ belongs to the label space $\mathcal{Y}_C$ with a cardinality of $C$. To simplify, we use one-hot encoding for the label space, where labels are represented by unit vectors in $\mathbb{R}^C$, and $\mathcal{Y}_C = \{e_c \mid c = 1, \ldots, C\}$ and $e_c$ is the $c$th standard unit vector in $\mathbb{R}^C$. We also use the notation $y = c$ instead of $y = e_c$. The class conditional distribution of $x$ given $y = c$ is denoted as $P_c(\cdot) := \mathbb{P}[x \in \cdot \mid y = c]$.

A classifier $h_W : \mathcal{X} \to \mathbb{R}^C$ assigns a *soft* label to an input point $x \in \mathcal{X}$, and its performance on the distribution $P$ is measured by the expected risk

$$L_P(h_W) := \mathbb{E}_{(x,y(x)) \sim P}[\ell(h_W(x), y(x))],$$

where $\ell : \mathbb{R}^C \times \mathcal{Y}_C \to [0, \infty)$ is a non-negative loss function (e.g., $L_2$ or cross-entropy losses).

We typically do not have direct access to the full population distribution $P$. Therefore, we generally aim to learn a classifier, $h$, using some balanced training data $S := \{(x_i, y_i)\}_{i=1}^m = \cup_{c=1}^C S_c = \cup_{c=1}^C \{x_{ci}, y_{ci}\}_{i=1}^{m_0} \sim P_B(m)$ of $m = C \cdot m_0$ samples consisting $m_0$ independent and identically distributed (i.i.d.) samples drawn from $P_c$ for each $c \in [C]$. Specifically, we intend to find $W$ that minimizes the regularized empirical risk

$$L_S^\lambda(h_W) := \frac{1}{m} \sum_{i=1}^m \ell(h_W(x_i), y_i) + \lambda \|W\|_2^2, \tag{1}$$

where the regularization controls the complexity of the function $h_W$ and typically helps reducing overfitting. Finally, the performance of the trained model is evaluated using the train and test error rates; $\mathrm{err}_S(h_W) := \frac{1}{m} \sum_{i=1}^m \mathbb{I}[\arg\max_c h_W(x_i)_c \neq y_i]$ and $\mathrm{err}_P(h_W) := \mathbb{E}_{(x,y) \sim P}[\mathbb{I}[\arg\max_c h_W(x)_c \neq y]]$, where $\mathbb{I} : \{\text{True}, \text{False}\} \to \{0, 1\}$ the indicator function.

**Neural networks.** In this work, the classifier $h_W$ is a neural network composed of a set of parametric layers. It is written as $h_W := e_{W_e} \circ f_{W_f}^L := e_{W_e} \circ g_{W_L}^L \circ \cdots \circ g_{W_1}^1$, where $g_{W_i}^i$ are parametric functions that map from $\mathbb{R}^{d_i}$ to $\mathbb{R}^{d_{i+1}}$, and $e_{W_e}$ is a linear function that maps from $\mathbb{R}^{d_{L+1}}$ to $\mathbb{R}^C$. These layers can be standard linear or convolutional layers (with ReLU activations) or a residual block. To simplify, we denote $f_i := g^i \circ \cdots \circ g^1$ and $h := h_W$. The specific architectures we used in the experiments are described in Appendix A.1.

**Optimization.** We optimize our models by minimizing the regularized empirical risk $L_S^\lambda(h)$ using Stochastic Gradient Descent (SGD) for a certain number of iterations $T$ with a regularization coefficient $\lambda > 0$. To do this, we initialize the weights $W_0$ of $h$ with a standard initialization distribution $Q$ of our choice (e.g., a standard normal distribution or the Kaiming He initialization (He et al., 2015)) and at each iteration, update $W_{t+1} \leftarrow W_t - \mu_t \nabla_W L_{\tilde{S}}(h_t)$, where $\mu_t > 0$ is the learning rate at the $t$-th iteration, and $\tilde{S} \subset S$ is a subset of size $B$ selected uniformly at random. Throughout the paper, we denote by $h_S^{W_t}$ the output of the learning

algorithm starting from $W_0$. When $W_0$ is not relevant or is obvious from the context, we will simply write $h_S^{W_0} = h_S = e_S \circ f_S$.

## 3 Neural Collapse and Generalization

In this section, we examine the theoretical connection between neural collapse and generalization. We begin by defining neural collapse, NCC separability, and effective depth of neural networks. We then explore how these concepts relate to the test-time performance of neural networks.

### 3.1 Nearest Class-Center Separability

Neural collapse (Papyan et al., 2020) identifies training dynamics of deep networks for standard classification tasks, in which the features of the penultimate layer associated with training samples belonging to the same class tend to concentrate around their class-means. This includes (NC1) class-features variability collapse, (NC2) the class means of the embeddings collapse to the vertices of a simplex equiangular tight frame, (NC3) the last-layer classifiers collapse to the class means up to scaling and (NC4) the classifier's decision collapses to simply choosing whichever class has the closest train class mean, while maintaining a zero classification error.

In this paper we focus on a weak form of NC4 we call *"nearest class-center separability"* (NCC separability). Formally, suppose we have a dataset $S = \cup_{c=1}^C S_c$ of samples and a mapping $f : \mathbb{R}^d \to \mathbb{R}^p$, the features of $f$ are NCC separable (w.r.t. $S$) if for all $i \in [m]$, we have $\hat{h}_f(x_i) = y_i$, where

$$\hat{h}_f(x) := \underset{c \in [C]}{\arg\min} \|f(x) - \mu_f(S_c)\|. \tag{2}$$

To measure the degree of NCC separability of a feature map $f$, we use the train and test classification error rates of the NCC classifier on top of the given layer, $\text{err}_S(\hat{h}_f)$ and $\text{err}_P(\hat{h}_f)$.

Essentially, NC4 asserts that during training, the feature embeddings in the penultimate layer become separable and the classifier $h$ itself converges to the 'nearest class-center classifier' $\hat{h}_f$.

### 3.2 Effective Depths and Generalization

In this section, we study the effective depths of neural networks and their connection with generalization. To formally define this notion, we focus on neural networks whose $L$ top-most layers are of the same size. We observe that neural networks trained for standard classification exhibit an implicit bias toward depth minimization.

**Observation 1** (Minimal depth hypothesis). *Suppose we have a dataset $S$. There exists an integer $L_0 \geq 1$, such that, if we properly train a neural network of any depth $L \geq L_0$ for cross-entropy minimization on $S$ using SGD with weight decay, the learned features $f^l$ become (approximately) NCC separable for all $l \in \{L_0, \ldots, L\}$.*

We note that if the $L_0$'th layer of $f_L$ exhibits NCC separability, we could correctly classify the samples already in the $L_0$'th layer of $f_L$ using a linear classifier (i.e., the nearest class-center classifier). Therefore, intuitively its depth is effectively upper bounded by $L_0$. The notion of effective depth of a neural network is formally defined as follows.

**Definition 1** ($\epsilon$-effective depth). *Suppose we have a dataset $S$ and a neural network $h = e \circ g^L \circ \cdots \circ g^1$ with $g^1 : \mathbb{R}^{d_1} \to \mathbb{R}^{d_2}$, $g^i : \mathbb{R}^{d_i} \to \mathbb{R}^{d_{i+1}}$ and linear classifier $e : \mathbb{R}^{d_{L+1}} \to \mathbb{R}^C$. Let $\hat{h}_i(x) := \arg\min_{c \in [C]} \|f_i(x) - \mu_{f_i}(S_c)\|$. The $\epsilon$-effective depth $\mathscr{d}_S^\epsilon(h)$ of the network $h$ is the minimal value $i \in [L]$, such that, $\text{err}_S(\hat{h}_i) \leq \epsilon$ (and $\mathscr{d}_S^\epsilon(h) = L$ if such $i \in [L]$ is non-existent).*

To avoid confusion, we note that the $\epsilon$-effective depth is a property of a neural network and not of the function it implements. That is, a function can be implemented by two different networks of different effective depths.

Informally, our observation (Obs. 1) suggests that when two sufficiently deep neural networks are properly trained (i.e., when the optimizer effectively minimizes the loss function), they will collapse to identical effective depths. However, achieving this result requires proper tuning of the hyperparameters to ensure that the neural network can be trained to minimize the loss function effectively.

While our empirical observations in Sec. 4 suggest that the optimizer learns neural networks of low depths, it is not necessarily the lowest depth that allows NCC separability. As a next step, we define the $\epsilon$-*minimal NCC depth*. Intuitively, the NCC depth of a given architecture is the minimal value $L \in \mathbb{N}$, for which there exists a neural network of depth $L$ whose features are NCC separable. As we will show, the relationship between the $\epsilon$-effective depth of a neural network and the $\epsilon$-minimal NCC depth is connected with generalization.

**Definition 2** ($\epsilon$-Minimal NCC depth)**.** *Suppose we have a dataset $S = \cup_{c=1}^{C} S_c$ and a neural network architecture $f^L = g^L \circ \cdots \circ g^1$ with $g^1 : \mathbb{R}^{d_1} \to \mathbb{R}^{d_0}$ and $g^i \in \mathcal{G} \subset \{g' \mid g' : \mathbb{R}^{d_0} \to \mathbb{R}^{d_0}\}$ for all $i = 2, \ldots, L$. The $\epsilon$-minimal NCC depth of $\mathcal{G}$ is the minimal depth $L$ for which there exist parameters $W = \{W_i\}_{i=1}^{L}$, such that, $f := f_W^L = g_{W_L}^L \circ \cdots \circ g_{W_1}^1$ satisfies $\mathrm{err}_S(\hat{h}_f) \leq \epsilon$, where $\hat{h}_f(x) := \arg\min_{c \in [C]} \|f(x) - \mu_f(S_c)\|$. We denote the $\epsilon$-minimal NCC depth by $\mathscr{d}_{\min}^{\epsilon}(\mathcal{G}, S)$.*

To study the performance of a given model, we consider the following setup. Let $S_1 = \{(x_i^1, y_i^1)\}_{i=1}^{m}$ and $S_2 = \{(x_i^2, y_i^2)\}_{j=1}^{m}$ be two balanced datasets. We consider them as two splits of the training dataset $S$, with the classifier $h_{S_1}^{W_0}$ representing the model selected by the learning algorithm using $S_1$, and $S_2$ being used to assess its performance. It is worth mentioning that although the model may differ slightly between training and testing, we regard $h_{S_1}^{W_0}$ as the function used during testing. For example, when the model incorporates Batch Normalization layers, we use mini-batch statistics during training. However, when evaluating the output model $h_{S_1}^{W_0}$ on new data, we switch to using population statistics. We denote by $X_j = \{x_i^j\}_{i=1}^{m}$ and $Y_j = \{y_i^j\}_{i=1}^{m}$ the instances and labels in $S_j$.

To formally state our bound, we make two technical assumptions. The first is that the misclassified labels that $h_{S_1}^{W_0}$ produces over the samples $X_2 = \cup_{c=1}^{C}\{x_{ci}^2\}_{i=1}^{m_0}$ are distributed uniformly.

**Definition 3** ($\delta_m$-uniform mistakes)**.** *We say that the mistakes of a learning algorithm $A : (S_1, W_0) \mapsto h_{S_1}^{W_0}$ are $\delta_m$-uniform, if with probability $\geq 1 - \delta_m$ over the selection of $S_1, S_2 \sim P_B(m)$, the values and indices of the mistaken labels of $h_{S_1}^{W_0}$ over $X_2$ are uniformly distributed (as a function of $W_0 \sim Q$).*

The above definition provides two conditions regarding the learning algorithm. It assumes that with a high probability (over the selection of $S_1, S_2$), $h_{S_1}^{W_0}$ makes the same number of mistakes on $S_2$ across all initializations $W_0$. In addition, it assumes that the mistakes are distributed uniformly across the samples in $S_2$ and their (incorrect) values are also distributed uniformly. While these assumptions may be violated in practice, the train error typically has a small variance and the mistakes are almost distributed uniformly when the classes are non-hierarchical (e.g., CIFAR10, MNIST).

For the second assumption, we consider the following term. Let $p \in (0, 1/2), \alpha \in (0, 1)$, we denote

$$\delta_{m,p,\alpha}^2 := \mathbb{P}_{S_1, S_2, \tilde{Y}_2, \hat{Y}_2}\left[\exists\, q \geq (1 + \alpha)\, p : \mathscr{d}_{\min}^{\epsilon}(\mathcal{G}, S_1 \cup \tilde{S}_2) > \mathbb{E}_{\hat{Y}_2}[\mathscr{d}_{\min}^{\epsilon}(\mathcal{G}, S_1 \cup \hat{S}_2)]\right]. \tag{3}$$

Here, $\tilde{Y}_2$ and $\hat{Y}_2$ denote sets of random labels. These labels are generated by first taking the set $Y_2$, then randomly selecting $\lfloor pm \rfloor$ ($\lfloor qm \rfloor$) labels from this set. Each of the selected labels is then replaced with a random selection from the set $[k]$. Let $\tilde{S}_2$ and $\hat{S}_2$ are datasets obtained by replacing the labels of $S_2$ with $\tilde{Y}_2$ and $\hat{Y}_2$ (resp.). We assume that $\delta_{m,p,\alpha}^2$ is small. Meaning, with a high probability, the minimal depth to fit $\lceil (2 - p)m \rceil$ correct labels and $\lfloor pm \rfloor$ random labels is upper bounded by the expected minimal depth to fit $\lceil (2 - q)m \rceil$ correct labels and $\lfloor qm \rfloor$ random labels for any $q \geq (1 + \alpha)p$.

While explicitly characterizing the values of $\delta_{m,p,\alpha}^2$ can be challenging, it is reasonable to assume that $\delta_{m,p,\alpha}^2$ is small since it is necessary to increase the model's capacity when fitting larger amounts of random labels (refer to Figs. 3). In order words, in both cases, the model has to simultaneously fit a significant amount of correct labels and at the same time $\lfloor pm \rfloor$ (or $\lfloor qm \rfloor$) random labels. Assuming that the target function $y$ is realizable by a sufficiently deep network, as $m$ increases, we can expect that the "majority" of the complexity required to fit the data would be devoted to fitting the random labels. This is because fitting these labels

typically demands an increase in network complexity while the target function requires a finite number of layers to implement. Therefore, as $m$ grows large, we anticipate that the inequality follows primarily from the need for deeper networks to fit more noisy labels.

Although proving this assumption in its entirety is difficult, the following proposition provides some intuition by demonstrating that the expected minimal depth required to fit a given dataset with random labels tends to infinity when increasing the number of training samples.

**Proposition 1.** *Let $d_0, m, L \geq 3$ and let $X = \{x_i\}_{i=1}^m \subset \mathbb{R}^{d_0}$ be a set of $m$ unlabeled samples. Let $\mathcal{G} = \{\sigma(Wx + b) \mid W \in \mathbb{R}^{d_0 \times d_0}, b \in \mathbb{R}^{d_0}\}$, where $\sigma$ is the ReLU element-wise activation function. Let $\tilde{Y} = (\tilde{y}_1, \ldots, \tilde{y}_m)$ be a set of $m$ uniformly distributed labels in $\{\pm 1\}$ and $\tilde{S} = \{(x_i, \tilde{y}_i)\}_{i=1}^m$. Then,*

$$\mathbb{E}_{\tilde{Y}}[\mathscr{d}_{\min}^\epsilon(\mathcal{G}, \tilde{S})] \geq \frac{C\sqrt{m} \cdot (1 - 2\epsilon)^2}{d_0 \log(m)\sqrt{\log(d_0)\log(em)}} \tag{4}$$

*for some universal constant $C > 0$.*

In the above proposition, we consider a binary classification problem (with labels $\{\pm 1\}$) and lower bound the expected number of fully-connected layers necessary to fit $m$ arbitrary training samples with random labels. The proposition shows that the expected value of the minimal depth, $\mathscr{d}_{\min}^\epsilon(\mathcal{G}, \tilde{S})$, necessary to fit a dataset with random labels increases as the number of training samples grows. Specifically, when fixing the width $d_0$ and threshold $\epsilon$, the right-hand side of the equation grows in proportion to $\sqrt{m}$, implying that $\mathbb{E}_{\tilde{Y}}[\mathscr{d}_{\min}^\epsilon(\mathcal{G}, \tilde{S})]$ must also grow accordingly.

Based on the previous proposition, we can further establish that the 0-minimal NCC depth required to fit a set of samples following the form described in equation 3 is expected to increase with the proportion $p$ of noisy labels in the data.

**Proposition 2.** *Let $d_0, m, L \geq 3$, and let $S_1, S_2 \subset \mathbb{R}^d \times \{\pm 1\}$ be two sets of $m$ labeled samples. Let $\tilde{Y}_2$ be labels that are generated by first taking the set $Y_2$ (the labels of $S_2$), then randomly selecting $\lfloor pm \rfloor$ labels from this set. Each of the selected labels is then replaced with a random selection from the set $\{\pm 1\}$. Let $\tilde{S}_2$ be the same set as $S_2$ with the labels $Y_2$ replaced with $\tilde{Y}_2$. Then,*

$$\mathbb{E}_{\tilde{Y}_2}[d_{\min}^0(\mathcal{G}, S_1 \cup \tilde{S}_2)] \geq \frac{C\sqrt{\lfloor pm \rfloor}}{d_0 \log(\lfloor pm \rfloor)\sqrt{\log(d_0)\log(e\lfloor pm \rfloor)}} \tag{5}$$

*for some universal constant $C > 0$.*

**Our generalization bound.** Following the above, we are ready to formulate our generalization bound.

**Proposition 3.** *Let $m \in \mathbb{N}$, $p \in (0, 1/2)$, $\alpha \in (0, 1)$ and $\epsilon \in (0, 1)$. Assume that the error of the learning algorithm is $\delta_m^1$-uniform. Assume that $S_1, S_2 \sim P_B(m)$. Let $h_{S_1}^{W_0}$ be the output of the learning algorithm given access to a dataset $S_1$ and initialization $W_0$. Then,*

$$\mathbb{E}_{S_1}\mathbb{E}_{W_0 \sim Q}[\text{err}_P(h_{S_1}^{W_0})] \leq \mathbb{P}_{S_1, S_2, \tilde{Y}_2}\left[\mathbb{E}_{W_0 \sim Q}[\mathscr{d}_{S_1}^\epsilon(h_{S_1}^{W_0})] \geq \mathscr{d}_{\min}^\epsilon(\mathcal{G}, S_1 \cup \tilde{S}_2)\right] \\ + (1 + \alpha)p + \delta_m^1 + \delta_{m,p,\alpha}^2, \tag{6}$$

*where $\tilde{Y}_2 = \{\tilde{y}_i\}_{i=1}^m$ denotes a set of random labels that are generated by first taking the set $Y_2$, then randomly selecting $\lfloor pm \rfloor$ labels from this set, each of the selected labels is re-selected randomly from the set $[k]$.*

The above proposition provides an upper bound on the expected test error of the classifier $h_{S_1}^{W_0}$ which is the term that we would like to bound. The proposition assumes that the mistakes $h_{S_1}^{W_0}$ generates on $X_2$ are distributed uniformly (with probability $\geq 1 - \delta_m^1$). To account the likelihood that this assumption fails, our bound includes the term $\delta_m^1$, which is assumed to be small.

Informally, the bound suggests the following idea to evaluate the performance of $h_{S_1}^{W_0}$. We start with an initial guess $p_m = p \in (0, 1/2)$ of the test error of $h_{S_1}^{W_0}$. Using this guess, we compare its $\epsilon$-effective depth with the minimal depth $\mathscr{d}_{\min}^\epsilon(\mathcal{G}, S_1 \cup \tilde{S}_2)$ required to NCC separate the samples in $S_1 \cup \tilde{S}_2$, where $\tilde{S}_2$ is the

result of randomly relabeling $p_m m$ of $S_2$'s labels. Intuitively, if the mistakes of $h_{S_1}^{W_0}$ are uniformly distributed and its $\epsilon$-effective depth is smaller than $\ell_{\min}^\epsilon(\mathcal{G}, S_1 \cup \tilde{S}_2)$, then, we expect $h_{S_1}^{W_0}$ to make at most $p_m$ mistakes on $S_2$. In a sense, the choice of $p_m$ serves as a 'guess' whether the effective depth of a model trained with $S_1$ is likely to be smaller than the minimal depth required to NCC separate the samples in $S_1 \cup \tilde{S}_2$.

Next, we interpret each term separately. The term $\mathbb{E}_{W_0 \sim Q}[\ell_{S_1}^\epsilon(h_{S_1}^{W_0})]$ depends on the complexity of the classification problem and the implicit bias of SGD to favor networks of small $\epsilon$-effective depths. In the worst case, if SGD does not minimize the $\epsilon$-effective depth or the labels in $S_1$ are random (and $m$ is sufficiently large), we expect $\mathbb{E}_{W_0 \sim Q}[\ell_{S_1}^\epsilon(h_{S_1}^{W_0})] = L$. On the other hand, $\ell_{\min}^\epsilon(\mathcal{G}, S_1 \cup \tilde{S}_2)$ measures the complexity of a task that involves fitting a dataset of size $2m$ samples, where $(2 - p_m)m \geq m$ of the labels are correct and $p_m m$ are random labels. By decreasing $p_m$, we expect $\ell_{\min}^\epsilon(\mathcal{G}, S_1 \cup \tilde{S}_2)$ to decrease, making the first term in the bound larger. In addition, if $h = e \circ f^L$ is a neural network of a fixed width, it is impossible to fit an increasing amount of random labels without increasing the depth. Therefore, when $p_m m \underset{m \to \infty}{\longrightarrow} \infty$, the dataset $S_1 \cup \tilde{S}_2$ becomes increasingly harder to fit, and we expect $\ell_{\min}^\epsilon(\mathcal{G}, S_1 \cup \tilde{S}_2)$ to tend to infinity. If $\mathbb{E}_{W_0 \sim Q}[\ell_{S_1}^\epsilon(h_{S_1}^{W_0})]$ is bounded as a function of $L$ and $m$ and if $p_m m \underset{m \to \infty}{\longrightarrow} \infty$, we obtain that $\mathbb{P}[\mathbb{E}_{W_0 \sim Q}[\ell_{S_1}^\epsilon(h_{S_1}^{W_0})] \geq \ell_{\min}^\epsilon(\mathcal{G}, S_1 \cup \tilde{S}_2)] \underset{m \to \infty}{\longrightarrow} 0$ and together with $p_m \underset{m \to \infty}{\longrightarrow} 0$, we have $\mathbb{E}_{S_1}[\text{err}_P(h_{S_1})] \leq \delta_m^1 + \delta_{m,p,\alpha}^2 + o_m(1)$.

As a side note, computing the expectation over $S_1, S_2$ in the bound is impossible, due to the limited access of the training data. However, instead, we empirically estimate this term using a set of $k$ pairs $(S_1^i, S_2^i)$ of $m$ samples, yielding an additional term that scales as $\mathcal{O}(1/\sqrt{k})$ to the bound (see Prop. 4 in the appendix).

Lastly, we would like to mention that the analysis presented above assumes a fixed width $d_0$ for each layer to simplify the analysis. However, it is straightforward to extend this analysis to cases where the width varies between layers in the trained network. The effective depth (Def. 1) of the trained model would still be measured as the first layer for which NCC separability occurs. The minimal complexity (Def. 2) would be the minimum number of layers with width $d_0$ required to achieve NCC separability up to an error of $\epsilon$, where $d_0$ represents the maximum width of a layer in the trained network.

## 3.3 Comparing Prop. 3 with Standard Generalization Bounds

Classic bounds (e.g., (Vapnik, 1998)) are based on bounding the test error with the sum between the train error together with a term $\mathcal{O}(\sqrt{\mathcal{C}(\mathcal{H})/m})$, where $\mathcal{C}(\mathcal{H})$ measures the complexity (e.g., VC dimension) of the class $\mathcal{H}$ (e.g., neural networks) and $m$ is the number of training samples. However, as discussed in Sec. 1, these bounds are vacuous in overparameterized learning regimes (e.g., training ResNet-50 on CIFAR10 classification). For instance, for VC-dimension based bounds (Vapnik, 1998), $\mathcal{C}(\mathcal{H})$ equals the VC-dimension of the class $\mathcal{H}$ which scales with the number of trainable parameters for ReLU networks (Bartlett et al., 2019). For example, even though the ResNet-50 architecture generalizes well when trained on CIFAR10, it has over 23 million parameters compared to the $m = 50000$ training samples in the dataset.

More recently, Neyshabur et al. (2015); Bartlett et al. (2017); Golowich et al. (2020); Neyshabur et al. (2018) suggested generalization bounds for neural networks that weakly depend on uniform convergence. In these bounds, the class-complexity $\mathcal{C}(\mathcal{H})$ is replaced with the individual complexity $\mathcal{C}(h_W)$ of the function we learn. For example, Golowich et al. (2020) proposed bounds that scale with $\mathcal{C}(h_W) = \rho^2 L$, where $L$ is the depth of $h_W$ and $\rho$ measures the product of the norms of its weight matrices. However, Nagarajan & Kolter (2019) showed that in certain cases unregularized least squares can generalize well even when its norm $\rho$ scales as $\Theta(\sqrt{m})$ and the bound becomes $\Theta_m(1)$. Furthermore, these bounds tend to be very large in practice (see Tab. 8 in (Neyshabur et al., 2019) and Tab. 1) and are negatively correlated with the test performance (Jiang et al., 2020). In addition, if the network's weight matrices' norms are larger than 1, quantities like $\rho$ grow exponentially when $L$ is varied. As shown in Tab. 1 this is empirically the case.

Our Prop. 3 offers a different way to measure generalization. Since this bound is not based on uniform convergence, it does not require that the network's complexity would be small in comparison to $m$; rather, the bound guarantees generalization if the network's effective size is smaller than that of a network that fits partially random labels. For instance, when the optimizer has a strong bias towards minimizing the effective

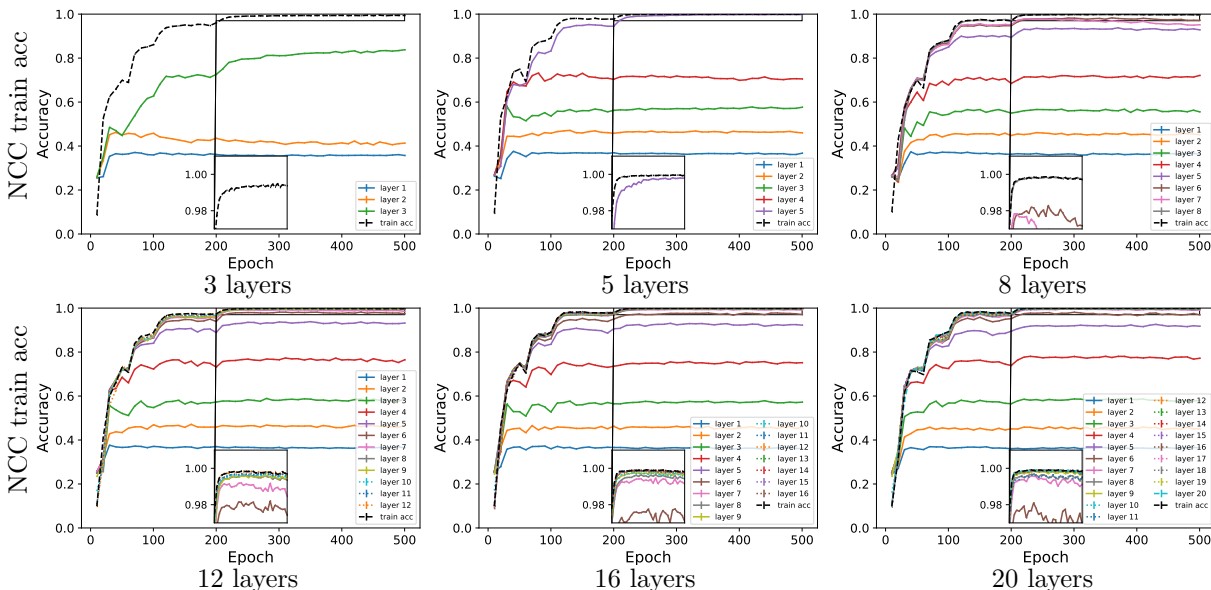

Figure 1: **Intermediate NCC separability of CONV-$L$-400 trained on CIFAR10.** We plot the NCC train accuracy rates of neural networks with varying numbers of layers. Each curve stands for a different layer within the network.

depth, $\mathbb{E}_{W_0 \sim Q}[\partial_{S_1}^{\epsilon}(h_{S_1}^{W_0})] \approx \partial_{\min}^{\epsilon}(\mathcal{G}, S_1)$ which is by definition upper bounded by $\partial_{\min}^{\epsilon}(\mathcal{G}, S_1 \cup \tilde{S}_2)$. We note that $\partial_{\min}^{\epsilon}(\mathcal{G}, S_1 \cup \tilde{S}_2)$ grows to infinity as $m \to \infty$ (since the network needs to memorize $m \to \infty$ random labels). On the other hand, $\partial_{\min}^{\epsilon}(\mathcal{G}, S_1)$ is bounded by the depth of a network that approximates the target function $y$ up to an approximation error $\epsilon$ (which typically exists due to universal approximation arguments). Therefore, for sufficiently large $m$, we expect to have $\partial_{\min}^{\epsilon}(\mathcal{G}, S_1 \cup \tilde{S}_2) > \partial_{\min}^{\epsilon}(\mathcal{G}, S_1)$. As we empirically see in Sec. 4, the effective depths of SGD-trained networks are usually small.

Unlike previous bounds, our bound has the advantage of being fairly independent of $L$. Namely, when the minimal depth hypothesis (Obs. 1) holds, we expect $\mathbb{E}_{W_0 \sim Q}[\partial_{S_1}^{\epsilon}(h_{S_1}^{W_0})]$ to be unaffected by the depth $L$ of $h_{S_1}^{W_0}$ (as long as $L \geq L_0$). Since $\partial_{\min}^{\epsilon}(\mathcal{G}, S_1 \cup \tilde{S}_2)$ is by definition independent of $L$, we expect $\mathbb{P}[\mathbb{E}_{W_0 \sim Q}[\partial_{S_1}^{\epsilon}(h_{S_1}^{W_0})] \geq \partial_{\min}^{\epsilon}(\mathcal{G}, S_1 \cup \tilde{S}_2)]$ to be independent of $L$ (when $L \geq L_0$). In Tab. 1 we empirically validate that our bound does not grow when increasing $L$.

## 4 Experiments

In this section, we experimentally analyze the emergence of neural collapse in the intermediate layers of neural networks. First, we validate the "Minimal Depth Hypothesis" (Obs. 1). Following that, we look at how corrupted labels affect the extent of intermediate layer NCC separability and the $\epsilon$-effective depth. We show that as the number of corrupted labels in the data increases, so does the $\epsilon$-effective depth. Finally, using the bound in Prop. 3, we provide non-trivial estimates of the test error. In Tab. 1, we empirically compare our bound with relevant baselines and show that, unlike other bounds, it achieves non-vacuous estimations of the test error. Throughout the experiments, we used Tesla-k80 GPUs for several hundred runs. Each run took between 5-20 hours. For additional experiments, see Appendix A. For clarity, in various plots we added a small rectangle located at the bottom of each plot to provide a magnified view of the lines between epochs 200 to 500, specifically within the y-axis range of 0.97 to 1.1. A vertical line connects the small rectangle with the zoomed-in area, while a horizontal line indicates the minimum value of y, which is set at y=0.97. The plots are high-definition pictures and are best viewed when zoomed in.

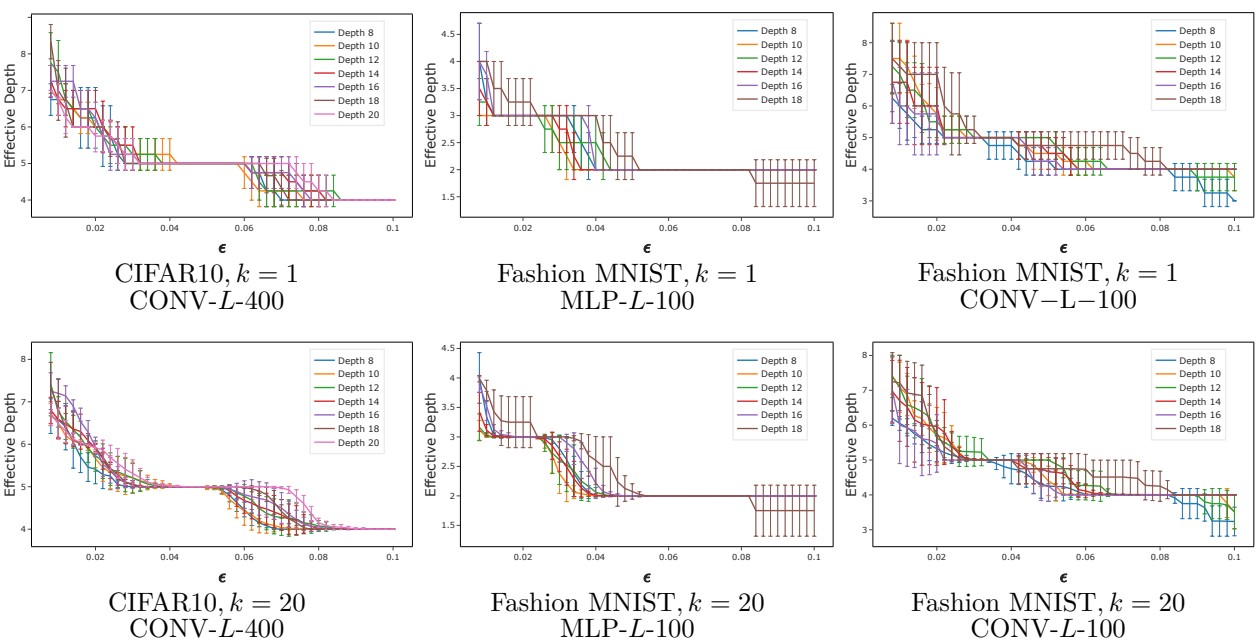

Figure 2: **Averaged $\epsilon$-effective depths over the last few epochs.** We plot the $\epsilon$-effective depth (y-axis) as a function of $\epsilon$ (x-axis). Each line specifies the $\epsilon$-effective depth of a neural network of a certain depth $L$. We show the averaged $\epsilon$-effective depth over the last $k = 1, 20$ epochs across 5 initializations. The network's architecture, dataset and $k$ are specified below each plot.

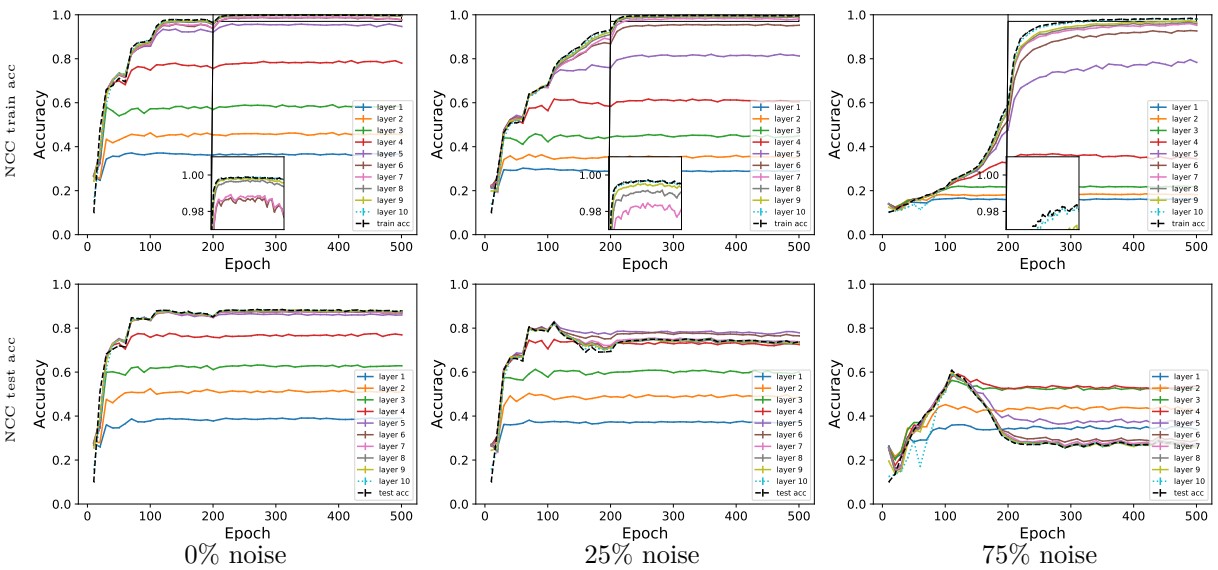

Figure 3: **Intermediate NCC separability of CONV-10-400 trained on CIFAR10 with partially corrupted labels.** We plot the NCC train/test accuracy rates of the various layers of a network trained with a certain amount of corrupted labels (see titles).

## 4.1 Setup

**Training process.** We consider $k$-class classification problems (e.g., CIFAR10) and train multilayered neural networks $h = e \circ f^L = e \circ g^L \circ \cdots \circ g^1 : \mathbb{R}^n \to \mathbb{R}^C$ on the corresponding training dataset $S$. The models are trained with SGD for cross-entropy loss minimization between its logits and the one-hot encodings of the

| Dataset | MNIST | | | Fashion MNIST | | | CIFAR10 | | | CIFAR10 | | |
|---|---|---|---|---|---|---|---|---|---|---|---|---|
| Architecture | CONV-10-50 | | | CONV-10-100 | | | CONV-16-100 | | | CONVRES-10-50 | | |
| $\mathbb{E}_{S_1, W_0}[\mathrm{err}_P(h_{S_1}^{W_0})]$ | 0.0075 | | | 0.0996 | | | 0.2676 | | | 0.29 | | |
| $p$ | 0.05 | 0.075 | 0.1 | 0.05 | 0.15 | 0.2 | 0.4 | 0.45 | 0.5 | 0.1 | 0.4 | 0.5 |
| Bound | 1.05 | 0.475 | **0.1** | 1.05 | 0.75 | **0.2** | **0.66** | 0.72 | 0.7 | **0.4** | **0.4** | 0.5 |

Table 2: Estimating the bound in Prop. 3. We used $\epsilon = 0.005$ to measure the effective depths.

| Dataset | MNIST | | | Fashion MNIST | | | CIFAR10 | | |
|---|---|---|---|---|---|---|---|---|---|
| | Arch | Test error | Bound | Arch | Test error | Bound | Arch | Test error | Bound |
| (Dziugaite & Roy, 2017)[1] | MLP-2-300 | 0.015 | 0.17 | - | - | - | - | - | - |
| (Biggs & Guedj, 2022)[2] | MLP-2-$H$ | 0.043 | 0.693 | MLP-2-$H$ | 0.153 | 0.976 | - | - | - |
| (Zhou et al., 2019)[3] | LeNet-5 | - | 0.46 | - | - | - | - | - | - |
| Ours | CONV-15-50 | 0.0074 | 0.1 | CONV-15-100 | 0.0996 | 0.2 | CONVRES-15-50 | 0.2804 | 0.4 |

Table 3: **Comparing our bound with baseline non-vacuous bounds in the literature.** Our error bound is reported in the fourth row, and the baseline bounds are reported in the bottom rectangle. We report the best bound achieved in each paper for each dataset. [1]The experiments were conducted on a binarized version of MNIST. [2]The specific width is not specified. [3]The test performance is not reported.

labels. We consistently use batch size 128, learning rate schedule with an initial learning rate 0.1, decayed three times by a factor of 0.1 at epochs 60, 120, and 160, momentum 0.9 and weight decay $5e-4$. Each model is trained for 500 epochs.

**Architectures.** We focused on three types of architectures: (a) MLP-$L$-$H$ with $L$ fully-connected layers of width $H$, (b) CONV-$L$-$H$ with $L$ $3 \times 3$ convolutional layers with padding 1, stride 1 and $H$ output channels and (c) a residual convolutional network CONVRES-$L$-$H$ with $L$ residual blocks with two $3 \times 3$ convolutional layers. In each network the layers are interlaced with batch normalization layers and ReLU activations. For more details see Appendix A.1.

**Datasets.** We consider various datasets: MNIST, Fashion MNIST, and CIFAR10. For CIFAR10 we used random cropping, random horizontal flips, and random rotations (by $15k$ degrees for $k$ uniformly sampled from [24]). All datasets were standardized.

## 4.2 Results

**Intermediate neural collapse.** To investigate the bias towards depth minimization, we trained several CONV-$L$-400 networks with varying depths on CIFAR10. Each plot in Fig. 1 illustrates the train NCC classification accuracy rates for every intermediate layer of a network of a specific depth. We made several interesting observations: **(i)** Networks with eight or more hidden layers display NCC train accuracy rates of about 100% in the eighth and higher layers, indicating that they are effectively of depth 7. **(ii)** The top layer embeddings become NCC separable at approximately the same epoch. **(iii)** The degree of NCC separability of intermediate layer $i$ converges as a function of $L$. In other words, the degree of NCC separability for each layer is more or less the same across all neural networks with a depth of at least 8, regardless of whether the layer is at the beginning or the end of the network.

For additional experiments and repeat results with various architectures and datasets, refer to Figs. 5-14 in Appendix A. In these experiments, we also report NCC train and test accuracy rates, along with additional measures of neural collapse when varying the depth. For instance, in Figs. 5 and 6, we present the outcomes with CONVRES-$L$-500.

**The effect of the depth on the $\epsilon$-effective depth.** As a next step we would like to verify Obs. 1. For this purpose, we would like to show that the $\epsilon$-effective depth remains largely consistent when two sufficiently deep neural networks are trained successfully. To validate this hypothesis we conducted the following experiments. We trained models on MNIST, Fashion MNIST and CIFAR10 with varying depth $L$. In Fig. 2 we plotted the averaged $\epsilon$-effective depths of each network's last $k = 1, 20$ epochs as a function of $\epsilon$. We also average the results across 5 different weight initializations and plot them along with error bar standard deviations. As can be seen, the $\epsilon$-effective depth is almost unaffected by the choice of $L$ for a

given $\epsilon$. Remarkably, for each $\epsilon$, the averaged effective depth varies very little across the various networks. Differently said, the $\epsilon$-effective depths of two trained deep networks of different depths are more or less the same, validating our Minimal Depth Hypothesis.

**NCC separability with partially corrupted labels.** Simply put, Prop. 3 compares the depths required to fit correct labels and partially corrupt labels. To better understand the effect of corrupted labels on the complexity of the task, we compare the $\epsilon$-effective depths of models trained with varying amounts of corrupted labels. Namely, we study the *degree* of NCC separability in the intermediate layers of neural networks that are trained with varying amounts of corrupted labels.

For this experiment, we trained instances of CONV-10-400 for CIFAR10 classification with 0%, 10% and 75% corrupted labels (e.g., uniformly distributed random labels). We plot the degrees of NCC separation on the train and test sets, $1 - \text{err}_S(\hat{h}_i)$ and $1 - \text{err}_P(\hat{h}_i)$, across the intermediate layers of the neural networks during the optimization procedure.

As can be seen in Fig. 3, when increasing the number of random labels, the degree of NCC separability across the intermediate layers tends to decrease. For example, when training with $\geq 25\%$ corrupted labels, the sixth layer's NCC accuracy rate drops lower than 98%, in comparison with training without corrupted labels that gives us $> 98\%$ accuracy. In particular, the $\epsilon$-effective depth of the former network is 6 while the latter's is 5 when $\epsilon = 0.02$ (see Def. 1). For additional experiments of this kind, see Figs. 15-19.

**Estimating the bound in equation 6.** We estimate the bound in equation 6 for multiple architectures and datasets. In each case, we used $\epsilon = 0.005$ by default and employed different 'guesses' $p$ (see Tab. 2) depending on the complexity of the learning task. We report an estimation of the expected test error of the models, $\mathbb{E}_{S_1,W_0}[\text{err}_P(h_{S_1}^{W_0})]$ and an estimation of the bound for each selection of $p$. For concrete technical details, see Appendix A.

As can be seen, for appropriate choices of $p$, we obtained non-trivial estimates of the test performance of the models, which is uncommon for standard bounds for deep neural networks. As expected, if the value of $p$ is too optimistic (e.g., close to $\mathbb{E}_{S_1,W_0}[\text{err}_P(h_{S_1}^{W_0})]$), then, the first term in the bound tends to be large compared to $\mathbb{E}_{S_1,W_0}[\text{err}_P(h_{S_1}^{W_0})]$. As predicted, when $p$ is increased, the first term in the bound decreases.

**Comparing our bound with standard generalization bounds.** We expect the bound in equation 6 to be insensitive to depth because the $\epsilon$-effective depth of deep neural networks is insensitive to depth, as shown in Fig. 2. We estimated the bound for various models and datasets, including CONV-$L$-50 trained on MNIST and CONV-$L$-100 trained on Fashion MNIST and CIFAR10, and CONVRES-$L$-50 trained on CIFAR10 with different values of $L$. The results, shown in Tab. 1, indicate that our bound gives similar values for each value of $L$. We also compared our bound to several norm-based generalization bounds for deep networks that can be found in (Bartlett & Mendelson, 2003; Neyshabur et al., 2015; Bartlett et al., 2017; Neyshabur et al., 2019) (we used the implementation of Neyshabur et al. (2019) to compute them). We found that our bound outperforms traditional bounds, as it is empirically non-vacuous and fairly independent of depth, while traditional bounds are extremely vacuous and rapidly increase with depth. These results support our prediction of the superiority of our bound over traditional bounds[1].

As mentioned in Section 1, recent papers (Biggs & Guedj, 2022; Dziugaite & Roy, 2017; Hellström & Durisi, 2022; Zhou et al., 2019) have obtained non-vacuous bounds in various learning settings. However, comparing our results with those of previous studies is challenging due to significant differences in theoretical and experimental setups.

On the theoretical side, Dziugaite & Roy (2017); Zhou et al. (2019); Hellström & Durisi (2022) used a PAC-Bayesian framework that incorporates random classifiers, while our paper focuses on standard training of deterministic neural networks using SGD. In (Biggs & Guedj, 2022), they derived generalization bounds for 2-layer networks, whereas our paper only considers deep networks. On the empirical side, the results of (Dziugaite & Roy, 2017) are based on a binary version of MNIST, where digits 0-4 are classified as 0, and digits 5-9 are classified as 1. Similarly, Hellström & Durisi (2022) experimented with MNIST, but only on a

---

[1]The norm-based generalization bounds could not be calculated for the CONVRES-$L$-50 architecture, as these bounds are not applicable for neural networks incorporating residual connections.

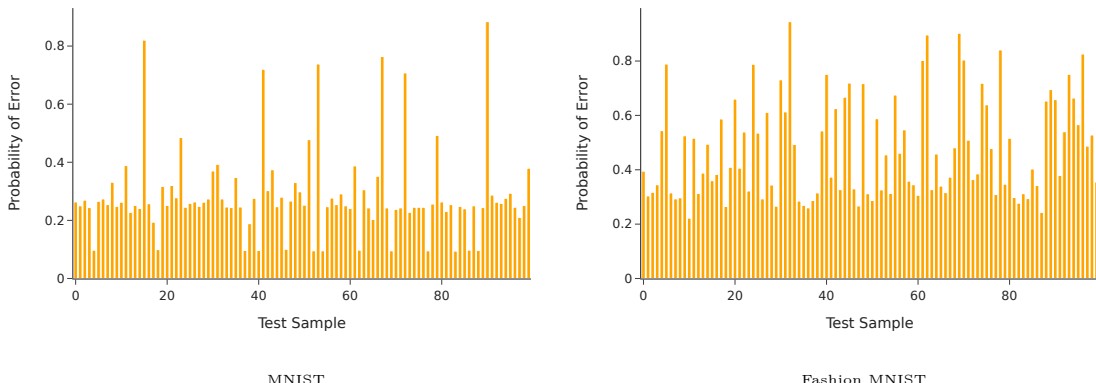

MNIST                    Fashion MNIST

Figure 4: The plot displays the probability of error across 100 random test samples, as we train 5000 networks with different initialization seeds, on 5000 randomly selected training samples for MNIST and Fashion MNIST datasets respectively.

binary version of MNIST that includes digits 4 and 9. In a different experiment, they evaluated their bound on a model trained on CIFAR10, but only after it was pre-trained on ImageNet, which could have influenced the test performance and the bound, particularly as CIFAR10 is a subset of ImageNet.

Given these variations, we compare some of the bounds by their best-performing setup reported in each paper in Tab. 3. For clarity, we provide disclaimers about the relaxations that each one makes. As shown in the table, our bound generally provides tighter bounds for neural networks.

**Validating mistakes uniformity.** To simplify the analysis in Section 3, we assumed that the mistakes made by the trained network on the test set are uniformly distributed across samples when the initialization is randomized. This assumption has two aspects: (1) the number of mistakes made by the network remains fixed across initializations, and (2) the probability of the network misclassifying a given test sample is unchanged across samples.

To validate this assumption empirically, we conducted multiple experiments. Firstly, we computed the standard deviations of the test errors of the models presented in Tab. 1. Since the standard deviations were found to be very low, it verifies that the number of mistakes on the test is more or less constant.

Additionally, we performed an experiment to validate uniformity. We trained 5000 CONV-10-50 networks on 5000 randomly selected samples from both the MNIST and Fashion MNIST datasets for 20 epochs each. For each test sample $(x_i, y_i)$, we measured the probability $p_i$ that it was misclassified by the model across the 5000 trained models. Then, we calculated the Coefficient of Variation $\eta = \sigma/p$ of $p_i$ across all samples, where $p$ and $\sigma$ represent the expectation and variance of $p_i$ across all samples, respectively. Essentially, $\eta$ measures the extent to which the error probability $p_i$ varies across samples when normalized against its mean. Our results showed that the Coefficient of Variation for this test was 0.5204 and 0.42 for MNIST and Fashion MNIST, respectively. We present the probability of error for 100 randomly selected test samples across the different networks in Fig. 4. Based on these experiments, it can be observed that the error probabilities are generally uniform, with only a minority of outlier samples that are difficult to classify accurately.

**Estimating $\delta^2_{m,p,\alpha}$.** We conducted an experiment where we took 10 pairs of splits $S^i_1$ and $S^i_2$, each with a size of $m = 25000$, and generated multiple corrupted labelings $\tilde{Y}^{i,p}_2$ for $S^i_2$, with corruption rates $p \in Q = \{q_1 = 0.01, q_2 = 0.05, q_3 = 0.1, q_4 = 0.15, q_5 = 0.20\}$. We needed to compute the minimal complexity required to fit a given dataset $S^{i,p}_3 := S^i_1 \cup \tilde{S}^{i,p}_2$, but doing so required significant computational power (as discussed in Appendix A). Instead, we estimated $\delta^2_{m,p,\alpha}$ by replacing $\ell^\epsilon_{\min}$ with $\ell^\epsilon_{S^{i,p}_3}$ and using trained CONV-10-50 models on $S^{i,p}_3$.

For each $q_j$, we trained a CONV-10-50 model $h_{i,q_j}$ on $S^{i,q_j}_3$, where $\tilde{S}^{i,q_j}_2$ is $S^i_2$ with its labels replaced by the sampled corrupted labels, with $q_j$ fraction being random. To estimate $\delta^2_{m,q_j,\alpha}$, for each $q_j \in Q$, we computed

the probability across $i$ that $\ell_{S_{i,3}^{q_j}}(h_{i,q_j})$ is larger than $\ell_{S_{i,3}^{q}}(h_{i,q})$ for some $q > q_j$, where $q \in Q$. We used $\alpha_j = q_{j+1}/q_j - 1$ for each $q_j$ since $(1+\alpha_j)q_j = q_{j+1}$. The following are the resulting estimates: $\delta_{m,0.01,4}^2 = 0$, $\delta_{m,0.05,1}^2 = 1/10$, $\delta_{m,0.1,0.5}^2 = 4/10$, $\delta_{m,0.15,1/3}^2 = 1$.

## 5 Conclusions

Understanding the ability of SGD to generalize well when training overparameterized neural networks is attributed as one of the major open problems in deep learning theory (Zhang et al., 2017). In this paper, we offer a new angle to study the role of depth in deep learning and the connection between neural collapse and generalization.

Our approach involves introducing the concept of effective depth, which identifies the lowest layer that exhibits NCC separability. We propose a novel generalization bound that estimates the likelihood that the effective depth of a trained neural network is strictly smaller than the minimal depth required to achieve NCC separability with partially corrupted labels. As demonstrated empirically, this criterion is a useful predictor of generalization. Furthermore, we characterize and empirically demonstrate that when sufficiently deep networks are trained, they converge to the same effective depth, implying that our bound is fairly constant when the depth is varied.

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

# A Additional Experiments and Details

## A.1 Architectures

In this section, we describe the architectures used in our experiments.

The first architecture is a convolutional network, denoted CONV-$L$-$H$, which consists of a stack of two $2 \times 2$ convolutional layers with stride 2, batch normalization, and ReLU activation. This is followed by $L$ stacks of blocks $g^i(x) = \sigma(B_i(C_i(x)))$, where $C_i$ is a $3 \times 3$ convolutional layer with $H$ channels, stride 1, and padding 1, $B_i$ is a batch normalization layer, and $\sigma$ is the ReLU activation. The final layer is linear. The $i$th intermediate layer refers to the output of the $i$th block of $g^i$.

The second architecture is an MLP, denoted MLP-$L$-$H$, which consists of $L$ hidden layers, where each layer $g^i(x) = \sigma(B_i(T_i(x)))$ contains a linear layer $T_i$ with output width $H$, followed by a batch normalization layer $B_i$ and a ReLU activation function $\sigma$. The final layer is linear.

The third architecture is a convolutional residual network, denoted CONVRES-$L$-$H$. It consists of a stack of two $2 \times 2$ convolutional layers with stride 2, batch normalization, and ReLU activation, followed by $L$ residual blocks. Each block computes $g^i(x) = \sigma(x + B_i^2(C_i^2(\sigma(B_i^1(C_i^1(x))))))$, where $C_i^j$ is a $3 \times 3$ convolutional layer with $H$ channels, stride 1, and padding 1, $B_i^j$ is a batch normalization layer, and $\sigma$ is the ReLU activation. The final layer is linear.

## A.2 Estimating the Generalization Bound

In this section we describe how we empirically estimate the bound in Prop. 3.

**Estimating the bound.** We would like to estimate the first term in the bound,

$$\mathbb{P}_{S_1, S_2, \tilde{Y}_2} \left[ \mathbb{E}_{W_0 \sim Q}[\ell_{S_1}^{\epsilon}(h_{S_1}^{W_0})] \geq \ell_{\min}^{\epsilon}(\mathcal{G}, S_1 \cup \tilde{S}_2) \right]. \tag{7}$$

According to Prop. 4 in order to estimate this term we need to generate i.i.d. triplets $(S_1^i, S_2^i, \tilde{Y}_2^i)$. Since we have limited access to training data, we use a variation of cross-validation and generate $k_1 = 5$ i.i.d. disjoint splits $(S_1^i, S_2^i)$ of the training data $S$. For each one of these pairs, we generate $k_2 = 3$ corrupted labeling $\tilde{Y}_2^{ij}$. We denote by $\tilde{S}_2^{ij}$ the set obtained by replacing the labels of $S_2^i$ with $\tilde{Y}_2^{ij}$ and $\tilde{S}_3^{ij} := S_1^i \cup \tilde{S}_2^{ij}$.

As a first step, we would like to estimate $\mathbb{E}_{W_0 \sim Q}[\ell_{S_1^i}^{\epsilon}(h_{S_1^i}^{W_0})]$ for each $i \in [k_1]$. For this purpose, we randomly select $T_1 = 5$ different initializations $W_0^1, \ldots, W_0^{T_1} \sim Q$ and for each one, we train the model $h_{S_1^i}^{W_0^t}$ using the training protocol described in Sec. 4.1. Once trained, we compute $\ell_{S_1^i}^{\epsilon}(h_{S_1^i}^{W_0^t})$ for each $t \in [T_1]$ (see Def. 1) and approximate $\mathbb{E}_{W_0 \sim Q}[\ell_{S_1^i}^{\epsilon}(h_{S_1^i}^{W_0})]$ using $d_i := \frac{1}{T_1} \sum_{t=1}^{T_1} \ell_{S_1^i}^{\epsilon}(h_{S_1^i}^{W_0^t})$.

As a next step, we would like to evaluate $\mathbb{I}[d_i \geq \ell_{\min}^{\epsilon}(\mathcal{G}, \tilde{S}_3^{ij})]$. We notice that $d_i \geq \ell_{\min}^{\epsilon}(\mathcal{G}, S_1^i \cup \tilde{S}_2^i)$ if and only if there is a $d_i$-layered neural network $f = g^{d_i} \circ \cdots \circ g^1$ for which $\mathrm{err}_{\tilde{S}_3^{ij}}(\hat{h}_f) \leq \epsilon$, where $\hat{h}_f(x) := \arg\min_{c \in [C]} \|f(x) - \mu_f(S_c)\|$. In general, computing this Boolean value is computationally hard. Therefore, to estimate this Boolean value, we simply train a $(d_i + 1)$-layered network $h = e \circ f$ and check whether its penultimate layer is $\epsilon$-NCC separable, i.e., $\mathrm{err}_{\tilde{S}_3^{ij}}(\hat{h}_f) \leq \epsilon$, where $\hat{h}_f(x) := \arg\min_{c \in [C]} \|f(x) - \mu_f(S_c)\|$. If SGD implicitly optimizes neural networks to maximize NCC separability as observed in (Papyan et al., 2020) (and also in this paper), we should expect to obtain $\epsilon$-NCC separability in the penultimate layer if that is possible with a $d_i$-layered network. Since training might be non-optimal, to obtain a robust estimation, we train $T_2 = 5$ models $h_t = e_t \circ f_t$ of depth $d_i + 1$ and pick the one with the best NCC separability in its penultimate layer. Namely, we replace $\ell_{\min}^{\epsilon}(\mathcal{G}, \tilde{S}_3^{ij})$ with $\min_{t \in [T_2]} \ell_{\tilde{S}_3^{ij}}^{\epsilon}(h_t)$ and estimate $\mathbb{I}[d_i \geq \ell_{\min}^{\epsilon}(\mathcal{G}, \tilde{S}_3^{ij})]$ using $\mathbb{I}[d_i \geq \min_{t \in [T_2]} \ell_{\tilde{S}_3^{ij}}^{\epsilon}(h_t)]$.

Our final estimation is the following

$$\frac{1}{k_1} \sum_{i=1}^{k_1} \frac{1}{k_2} \sum_{j=1}^{k_2} \mathbb{I}\left[d_i \geq \min_{t \in [T_2]} \mathscr{d}_{\tilde{S}_3^{ij}}^{\epsilon}(h_t)\right] \approx \mathbb{P}_{S_1, S_2, \tilde{Y}_2}\left[\mathbb{E}_{W_0 \sim Q}[\mathscr{d}_{S_1}^{\epsilon}(h_{S_1}^{W_0})] \geq \mathscr{d}_{\min}^{\epsilon}(\mathcal{G}, S_1 \cup \tilde{S}_2)\right]. \tag{8}$$

In order to estimate the bound we assume that $\delta_m^1$ and $\delta_{m,p,\alpha}^2$ are negligible constants and that $\alpha = 1$. The estimation of the bound is given by the sum of the left-hand side in equation 8 and $p$.

**Estimating the mean test error.** To estimate the mean test error, $\mathbb{E}_{S_1, W_0}[\mathrm{err}_P(h_{S_1}^{W_0})]$, as typically done in machine learning, we replace the population distribution $P$ with the test set $S_{test}$ and we replace the expectation over $S_1$ and $W_0$ with averages across the $k_1 = 5$ random selections of $\{S_1^i\}_{i=1}^{k_1}$ and $T_1 = 5$ random selections of $\{W_0^t\}_{t=1}^{T_1}$. Namely, we compute the following $\frac{1}{k_1} \sum_{i=1}^{k_1} \frac{1}{T_1} \sum_{t=1}^{T_1} \mathrm{err}_{S_{test}}(h_{S_1^i}^{W_0^t}) \approx \mathbb{E}_{S_1, W_0}[\mathrm{err}_P(h_{S_1}^{W_0})]$.

## A.3 Neural Collapse

To obtain a comprehensive analysis of collapse across layers, we also estimate the degree of NC1.

To evaluate NC1, we follow the process suggested by Galanti et al. (2022), which is a simplified version of the original approach of Papyan et al. (2020). For a feature map $f : \mathbb{R}^d \to \mathbb{R}^p$ and two (class-conditional) distributions[2] $Q_1, Q_2$ over $\mathcal{X} \subset \mathbb{R}^d$, we define their *class-distance normalized variance* (CDNV) to be

$$V_f(Q_1, Q_2) := \frac{\mathrm{Var}_f(Q_1) + \mathrm{Var}_f(Q_2)}{2\|\mu_f(Q_1) - \mu_f(Q_2)\|^2},$$

where $\mu_u(Q) := \mathbb{E}_{x \sim Q}[u(x)]$ and by $\mathrm{Var}_u(Q) := \mathbb{E}_{x \sim Q}[\|u(x) - \mu_u(Q)\|^2]$ the mean and variance of $u(x)$ for $x \sim Q$. Essentially, this quantity measures to what extent the feature vectors of samples from $Q_1$ and $Q_2$ are separated and clustered in space.

To demonstrate the gradual evolution of collapse across the layers, for each sub-architecture $f^i = g^i \circ \cdots \circ g^1(x)$ we consider the train and test class features variations $\mathrm{Avg}_{c \neq c'}[V_{f^i}(S_c, S_{c'})]$ and $\mathrm{Avg}_{c \neq c'}[V_{f^i}(P_c, P_{c'})]$. The population distribution of each class, $P_c$, is replaced with the test samples of that class.

As shown by Galanti et al. (2022), this definition is essentially the same as that of Papyan et al. (2020). Furthermore, they showed that the NCC classification error rate can be upper bounded in terms of the CDNV. However, the NCC error can be zero in cases where the CDNV is larger than zero. For example, if the two classes are uniformly distributed over the 1-radius circles around the points $(-1, 0)$ and $(1, 0)$ in $\mathbb{R}^2$, then they are perfectly NCC separable while the CDNV between the two distributions is 0.25.

**Auxiliary experiments on the effective depth.** In Figs. 7-14 we plot the NCC and the CDNV rates of neural networks with varying numbers of layers evaluated on the train and test data. Each curve stands for a different layer within the network. As can be seen, in all cases, for networks deeper than a threshold we obtain (near-perfect) NCC separability in all of the top layers. Furthermore, the degree of class-features variability collapse increases with the network's depth as depicted by decreasing CDNVs.

**Auxiliary experiments with noisy labels.** In Figs. 15-19 we repeat the experiment in Fig. 3 and plot the results of the same experiment, with different networks and datasets (see captions). As can be seen, the effective NCC depth of a neural network tends to increase as we train with increasing amounts of corrupted labels.

---

[2]The definition can be extended to finite sets $S_1, S_2 \subset \mathcal{X}$ by defining $V_f(S_1, S_2) = V_f(U[S_1], U[S_2])$.

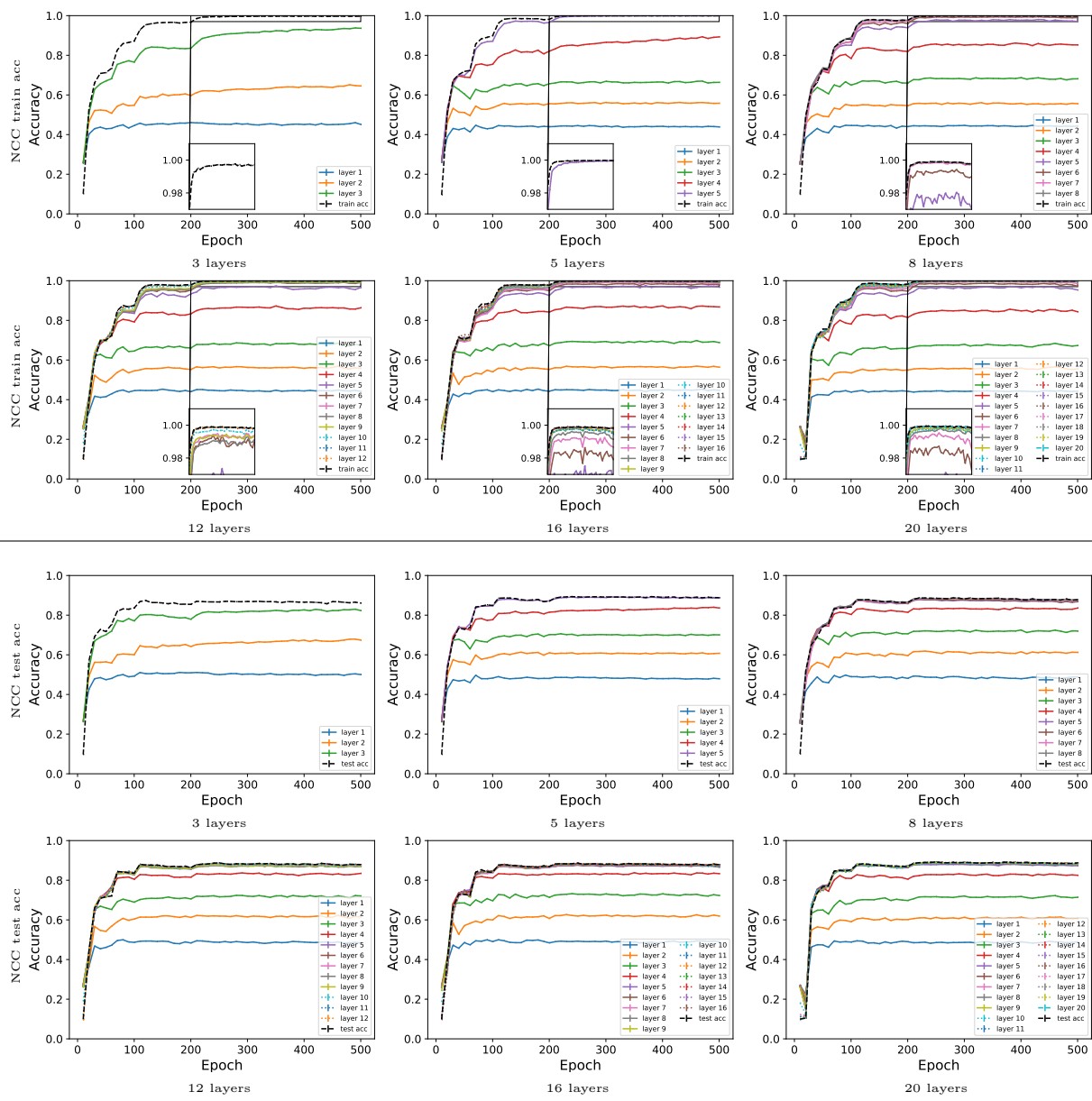

Figure 5: **Intermediate NCC separability of CONVRES-$L$-500 trained on CIFAR10.** We plot the NCC train and test accuracy rates of neural networks with varying numbers of layers. Each curve stands for a different layer within the network.

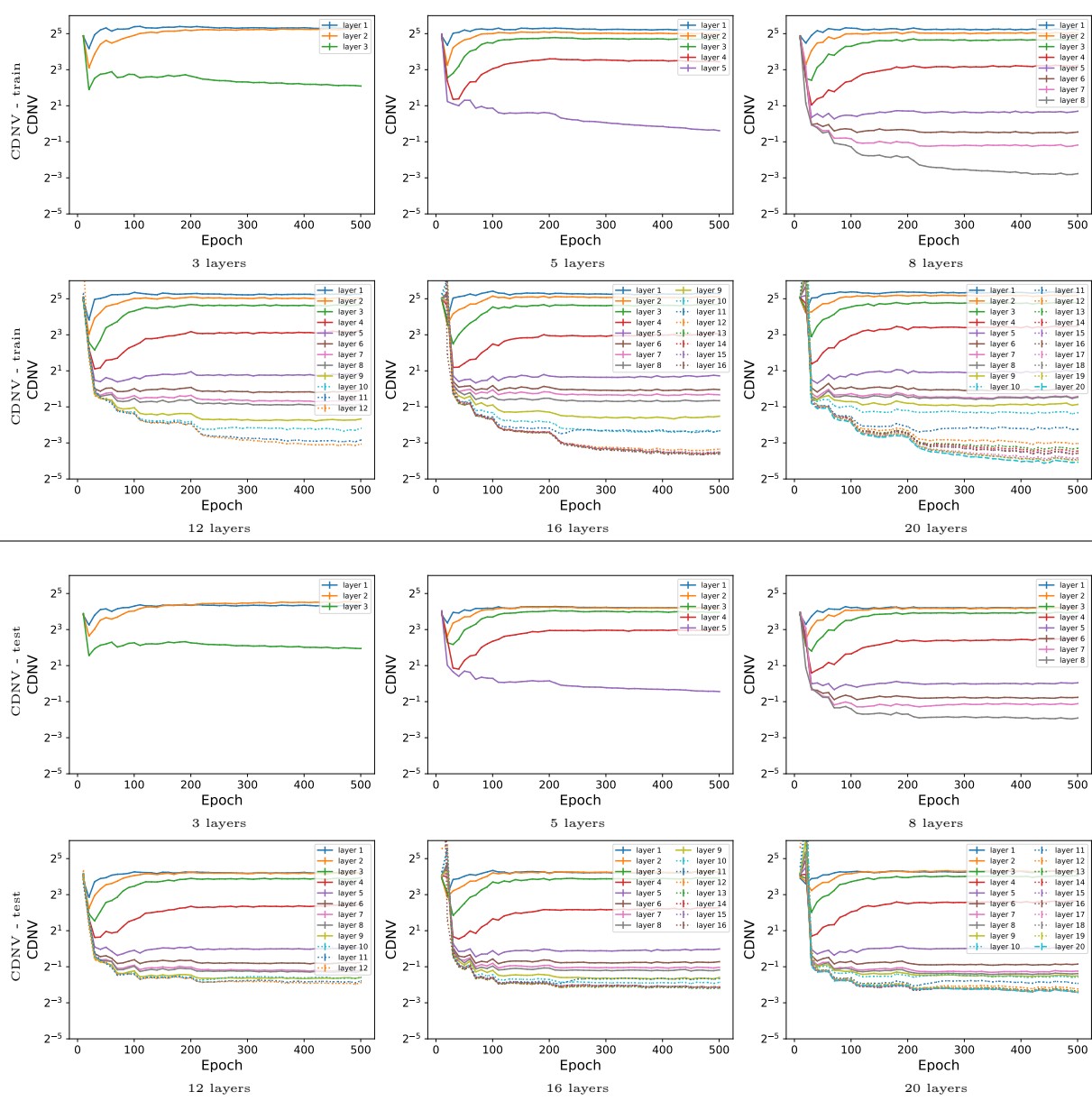

Figure 6: **Intermediate class-features variability collapse separability of CONVRES-$L$-500 trained on CIFAR10.** We plot the CDNV on the training and test data of neural networks with varying numbers of layers. Each curve stands for a different layer within the network.

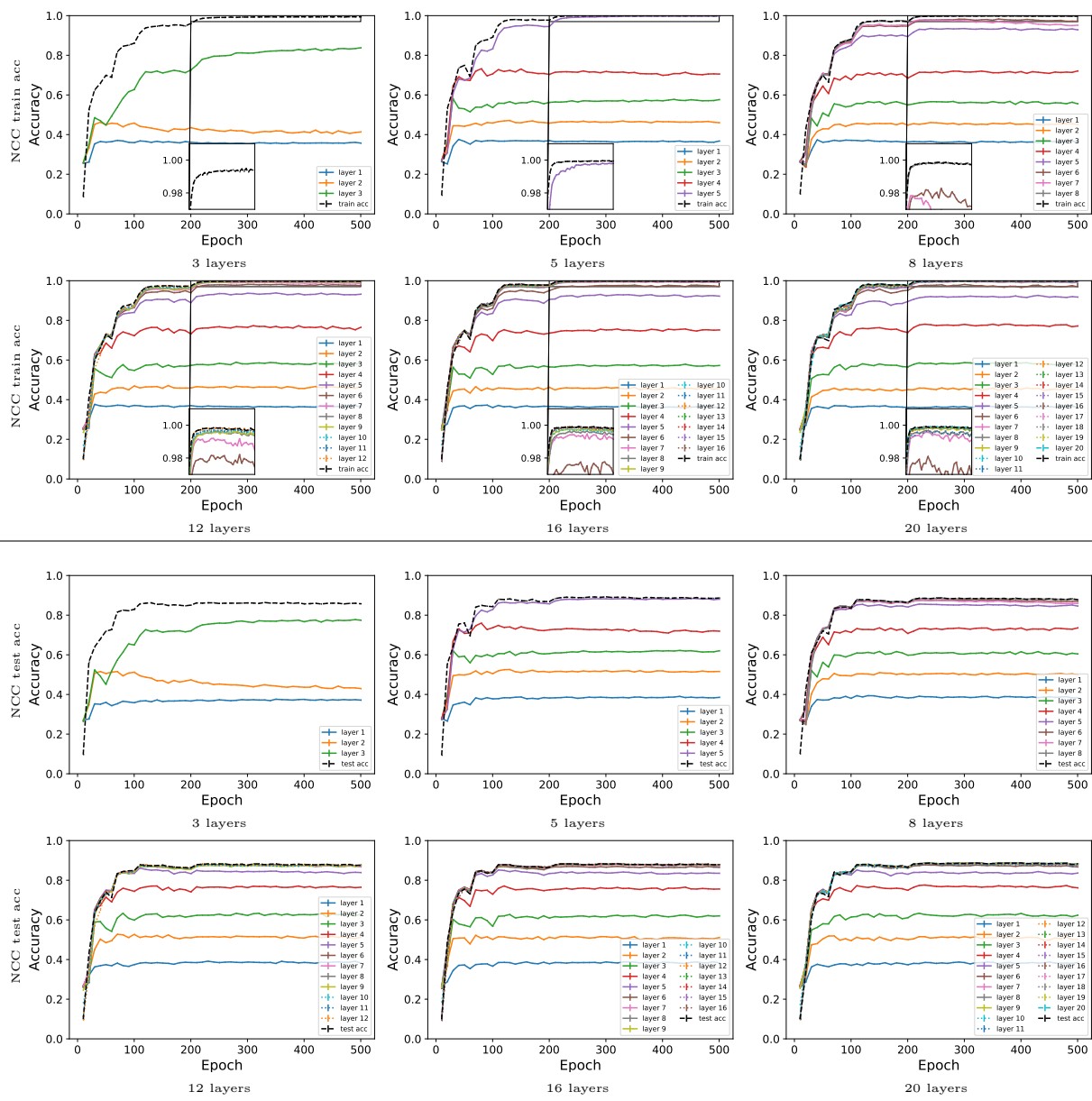

Figure 7: **Intermediate NCC separability of CONV-$L$-400 trained on CIFAR10.** We plot the NCC train and test accuracy rates of neural networks with varying numbers of layers. Each curve stands for a different layer within the network.

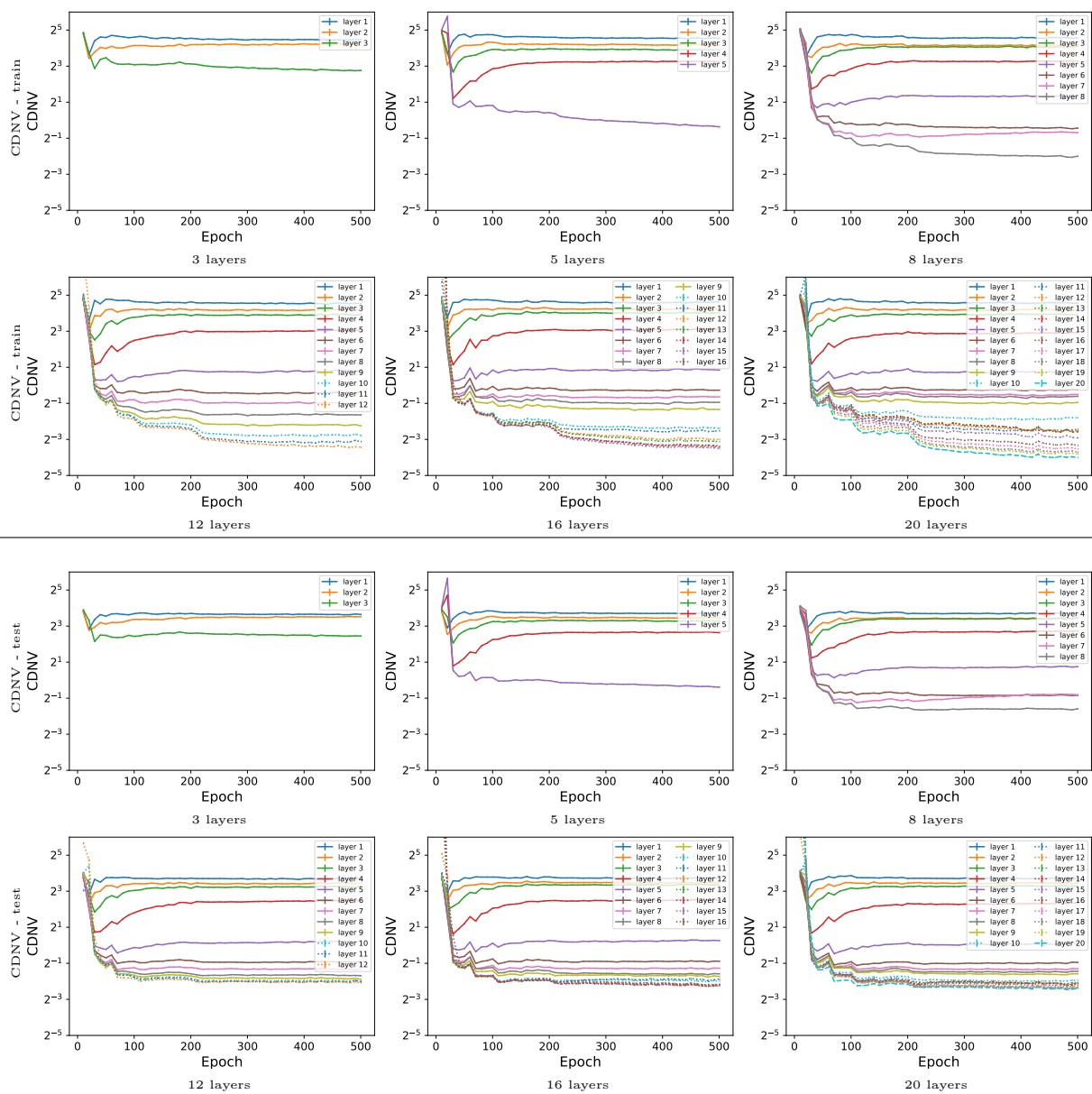

Figure 8: **Intermediate class-features variability collapse separability of CONV-$L$-400 trained on CIFAR10.** We plot the CDNV on the training and test data of neural networks with varying numbers of layers. Each curve stands for a different layer within the network.

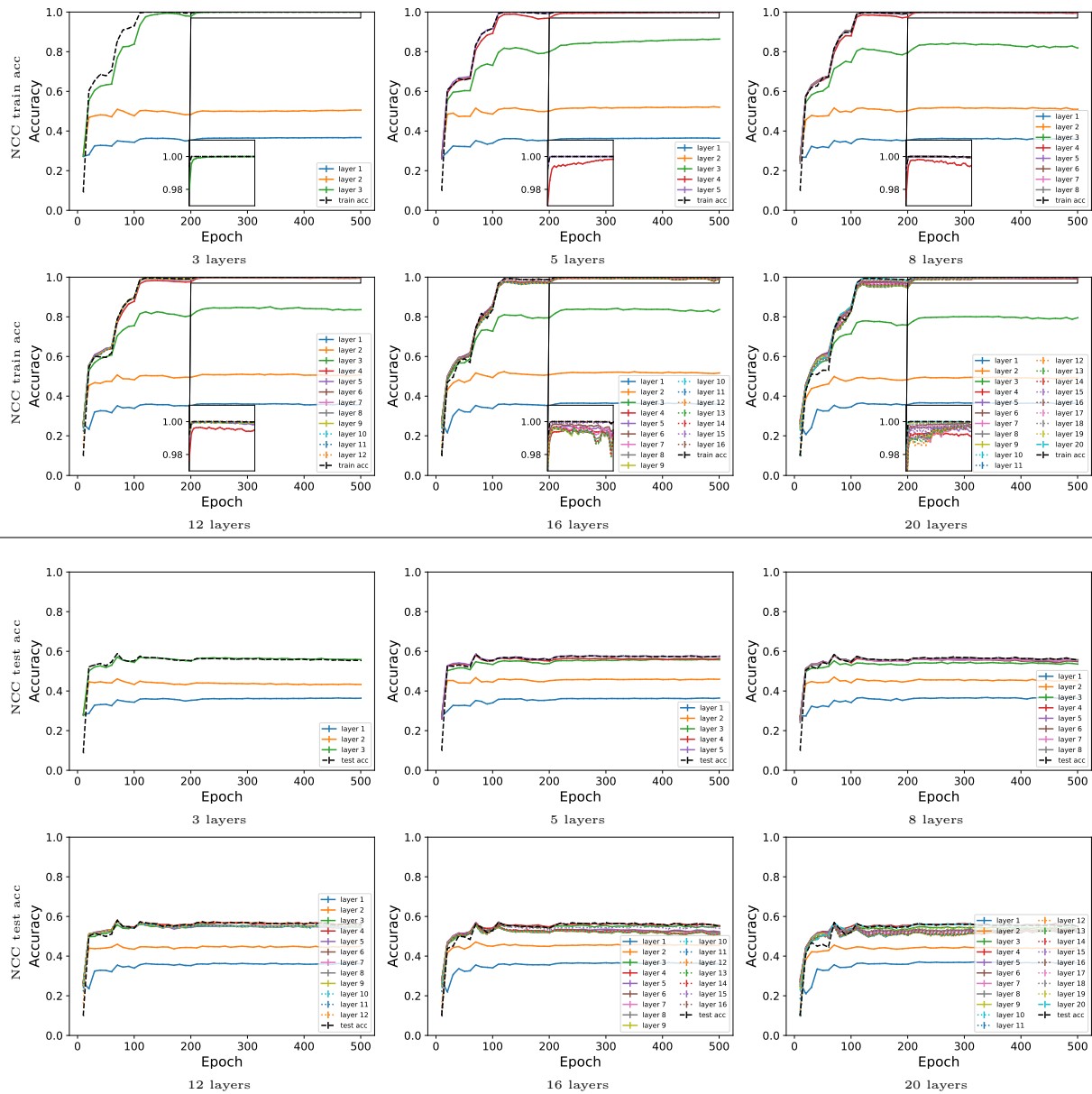

Figure 9: **Intermediate neural collapse of MLP-$L$-300 trained on CIFAR10.** We plot the NCC train and test accuracy rates of neural networks with varying numbers of layers. Each curve stands for a different layer within the network.

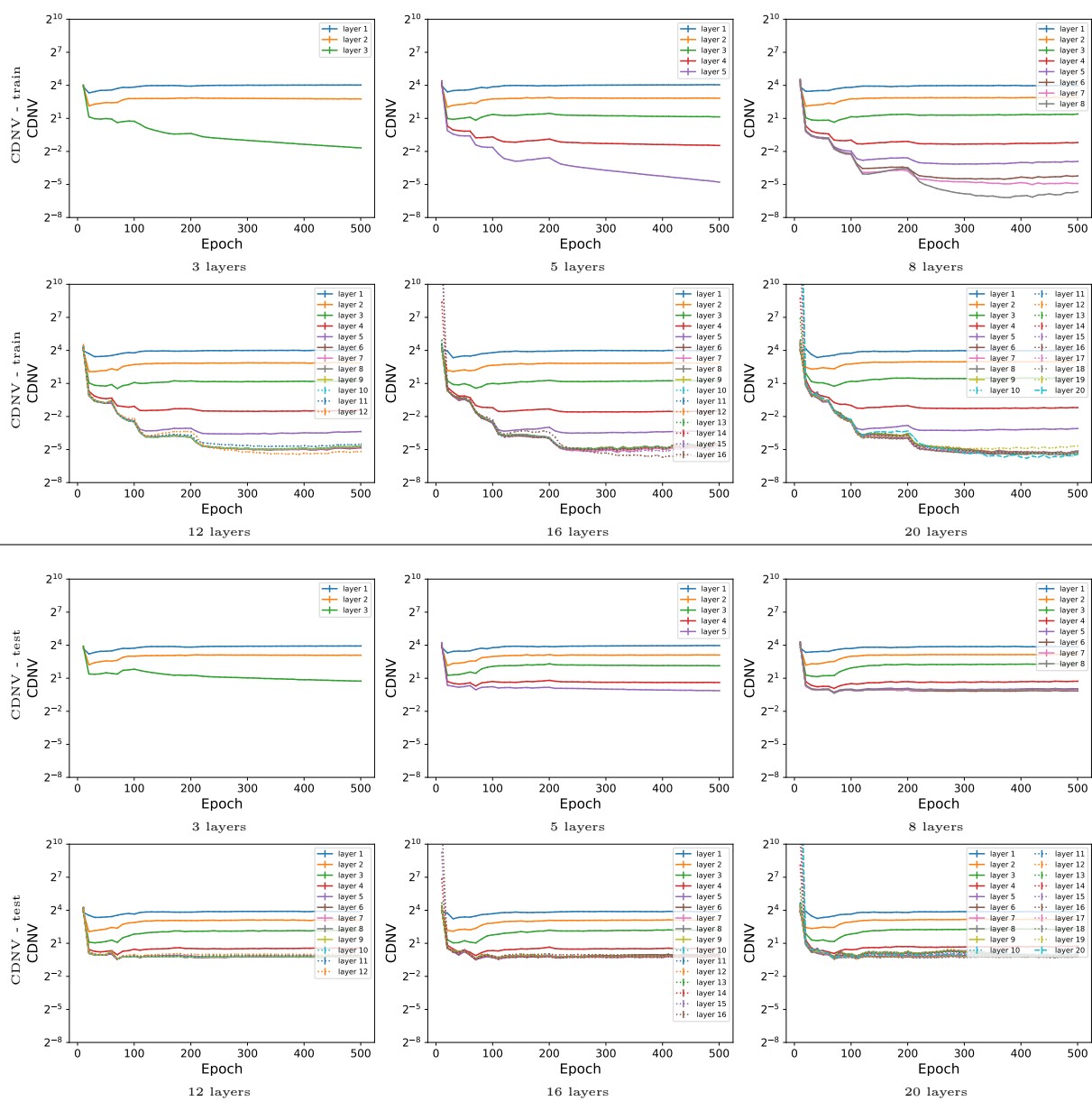

Figure 10: **Intermediate class-features variability collapse separability of MLP-$L$-300 trained on CIFAR10.** We plot the CDNV on the training and test data of neural networks with varying numbers of layers. Each curve stands for a different layer within the network.

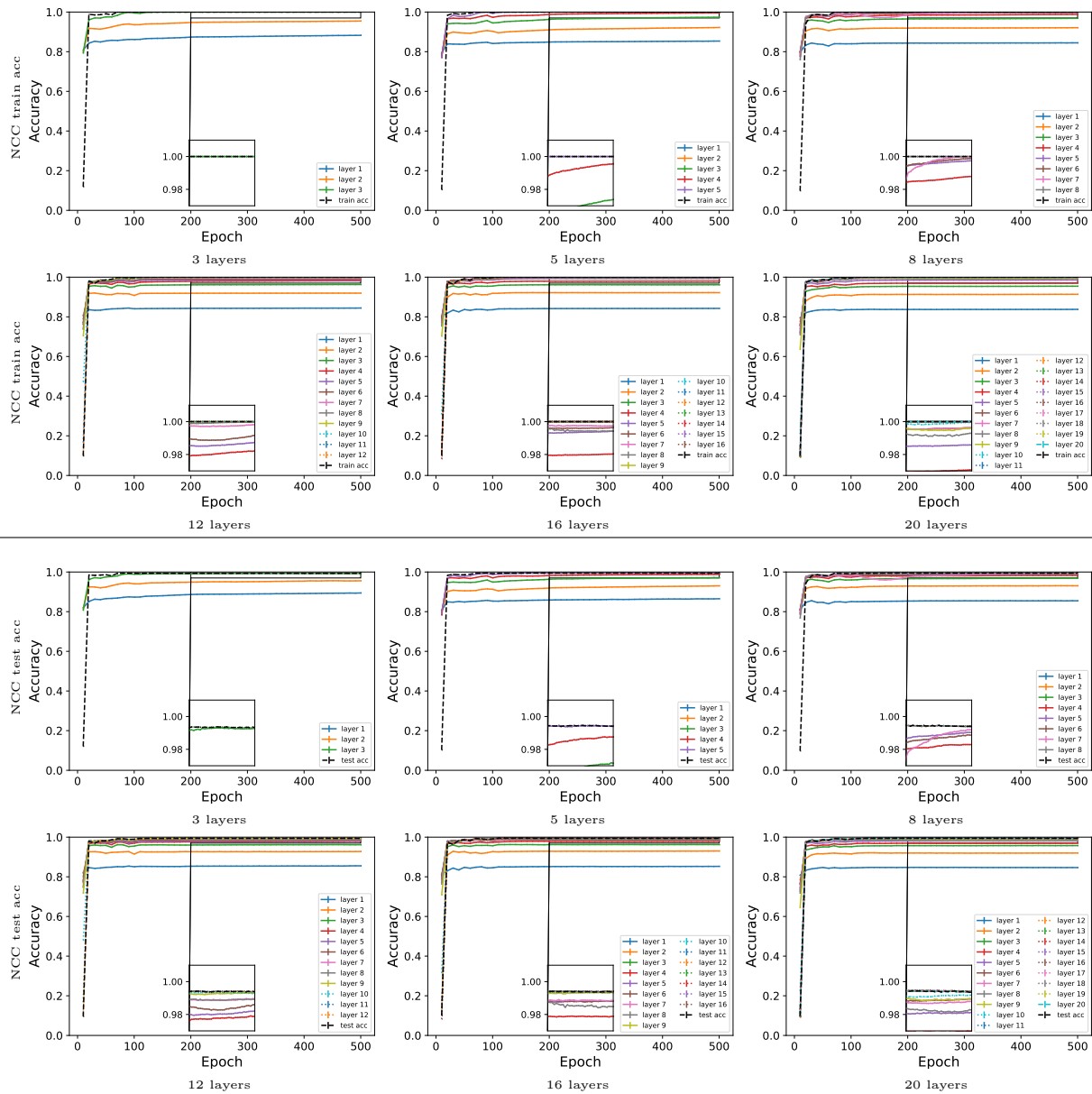

Figure 11: **Intermediate neural collapse of CONV-$L$-50 trained on MNIST.** We plot the NCC train and test accuracy rates of neural networks with varying numbers of layers. Each curve stands for a different layer within the network.

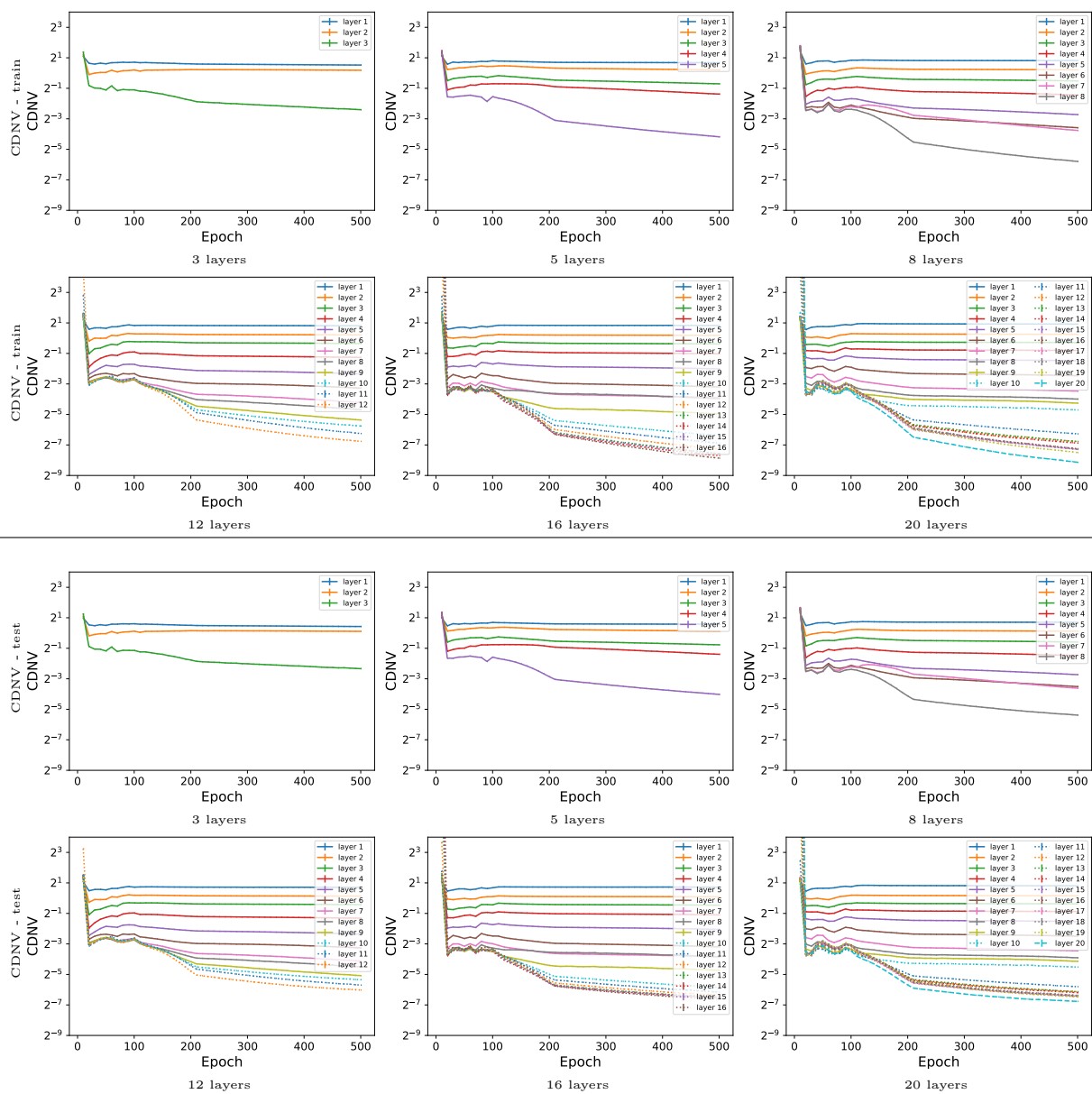

Figure 12: **Intermediate class-features variability collapse separability of CONV-$L$-50 trained on MNIST.** We plot the CDNV on the training and test data of neural networks with varying numbers of layers. Each curve stands for a different layer within the network.

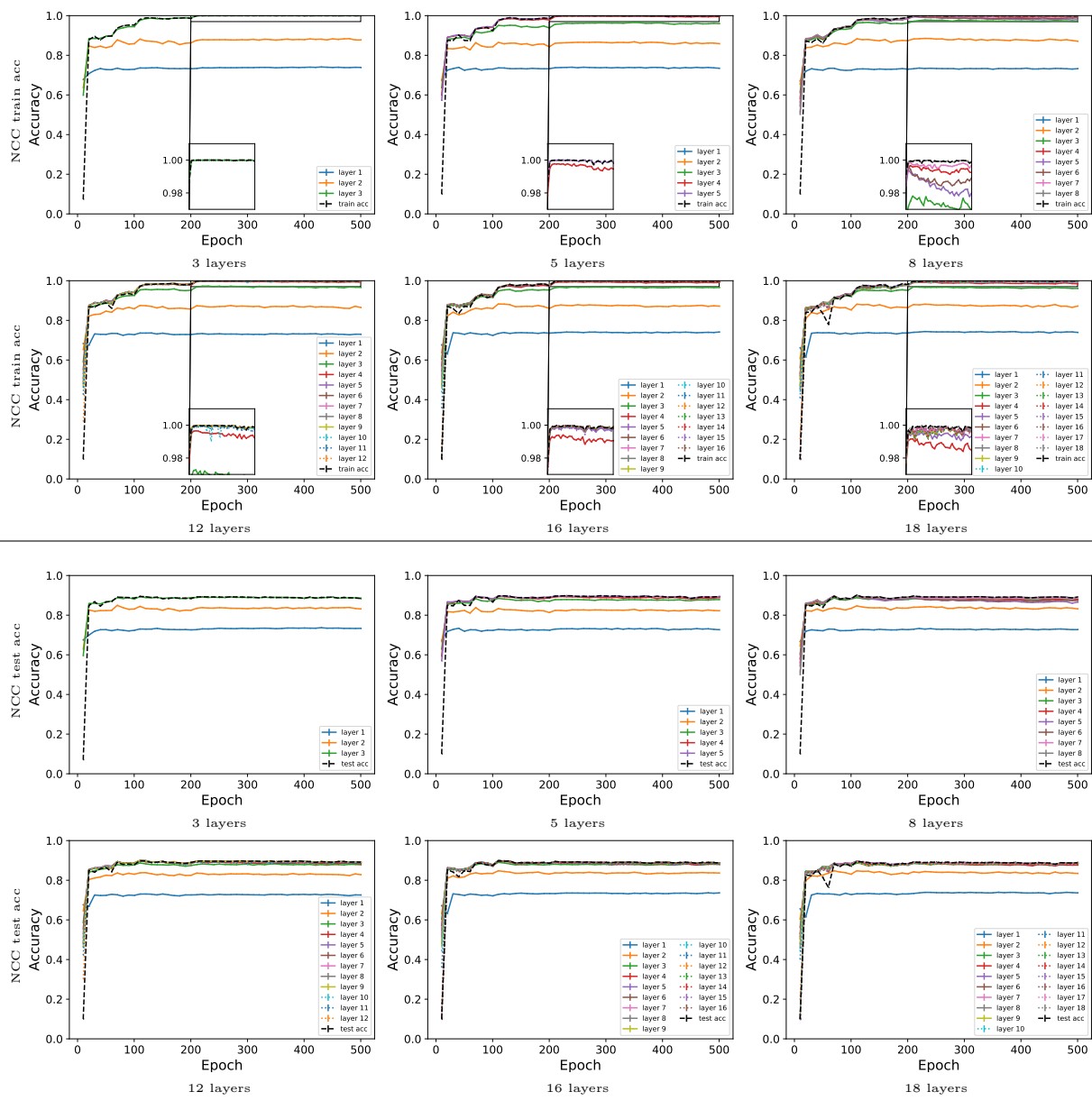

Figure 13: **Intermediate neural collapse of MLP-$L$-100 trained on Fashion MNIST.** We plot the NCC train and test accuracy rates of neural networks with varying numbers of layers. Each curve stands for a different layer within the network.

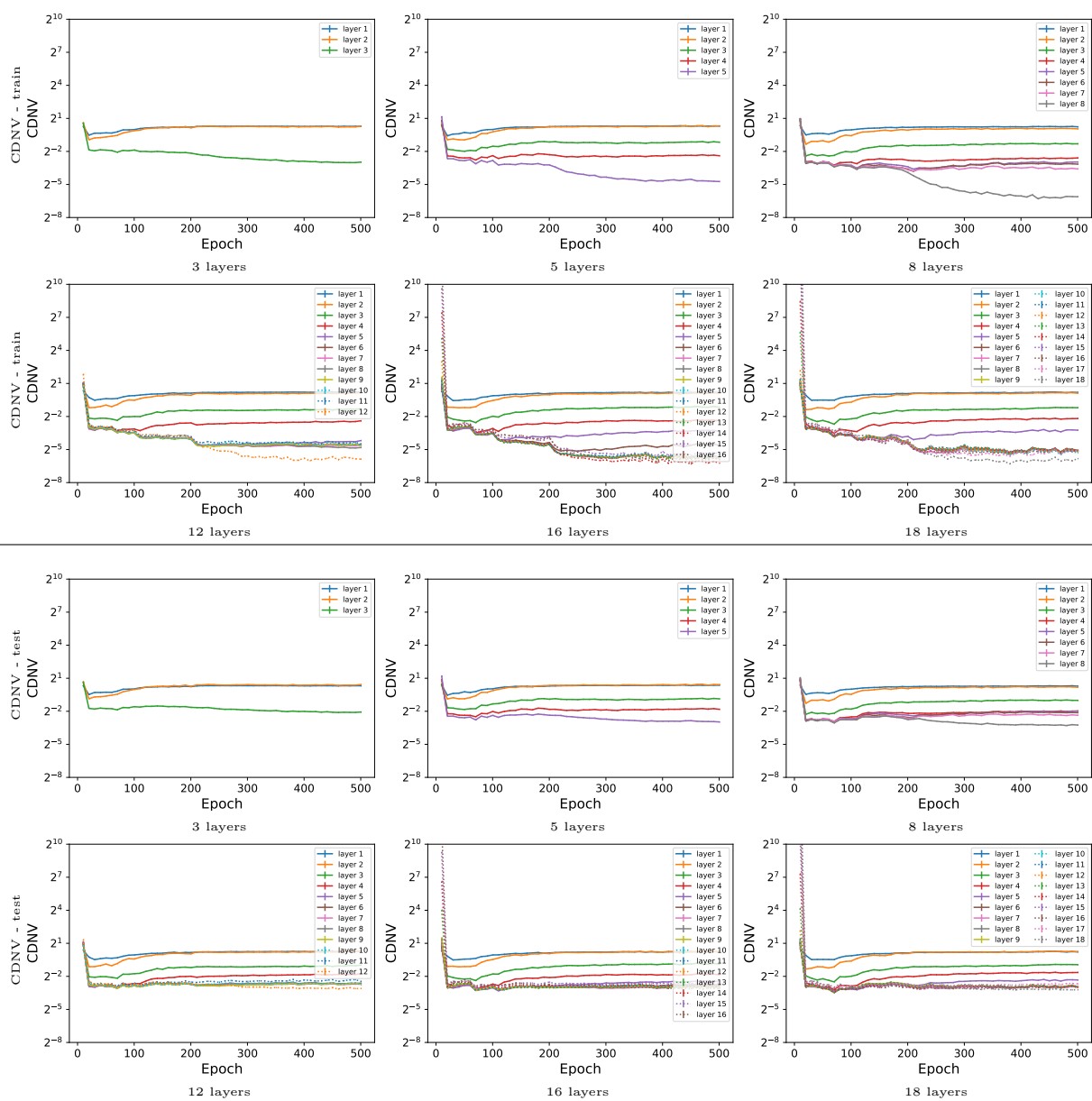

Figure 14: **Intermediate class-features variability collapse separability of MLP-$L$-100 trained on Fashion MNIST.** We plot the CDNV on the training and test data of neural networks with varying numbers of layers. Each curve stands for a different layer within the network.

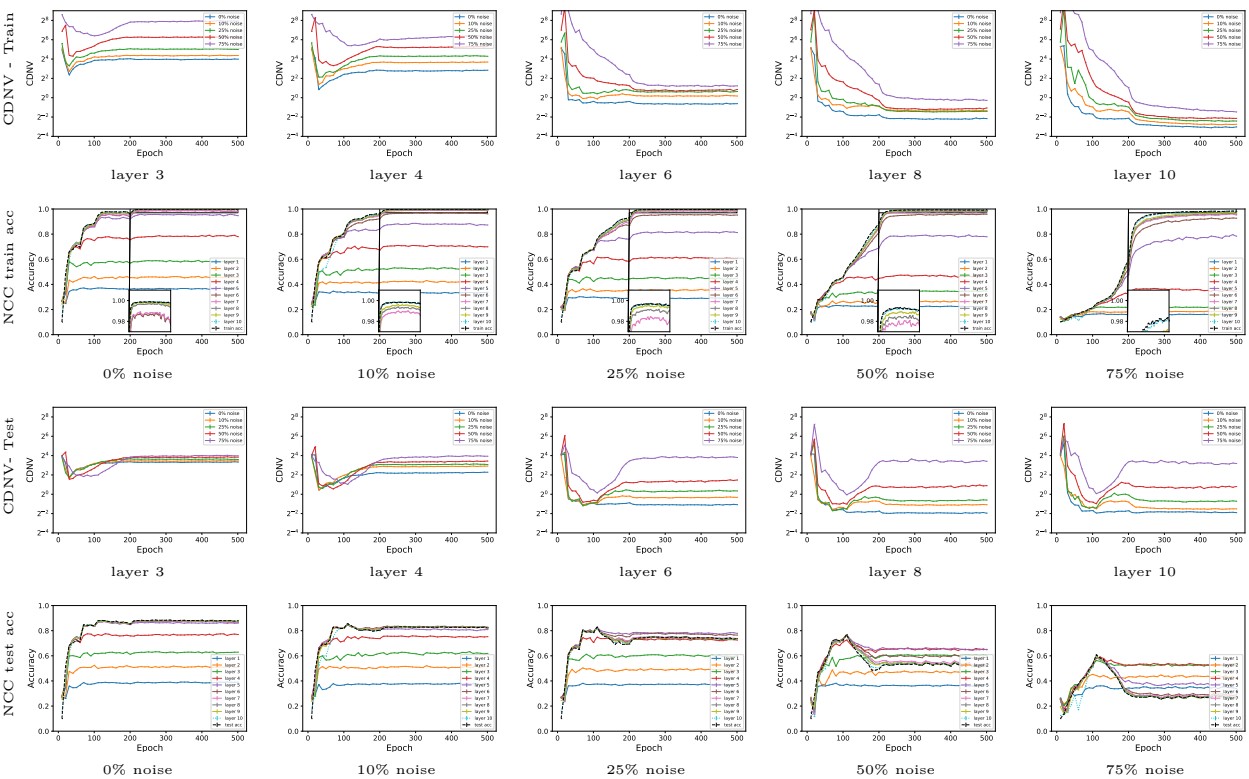

Figure 15: **Intermediate neural collapse of CONV-10-400 trained on CIFAR10 with partially corrupted labels.** In the first (third) row, we plot the CDNV on the train (test) data for intermediate layers of networks trained with varying amounts of corrupted labels (see legend). In the second (fourth) row, we plot the NCC accuracy rates of the various layers of a network trained with a certain amount of corrupted labels (see titles).

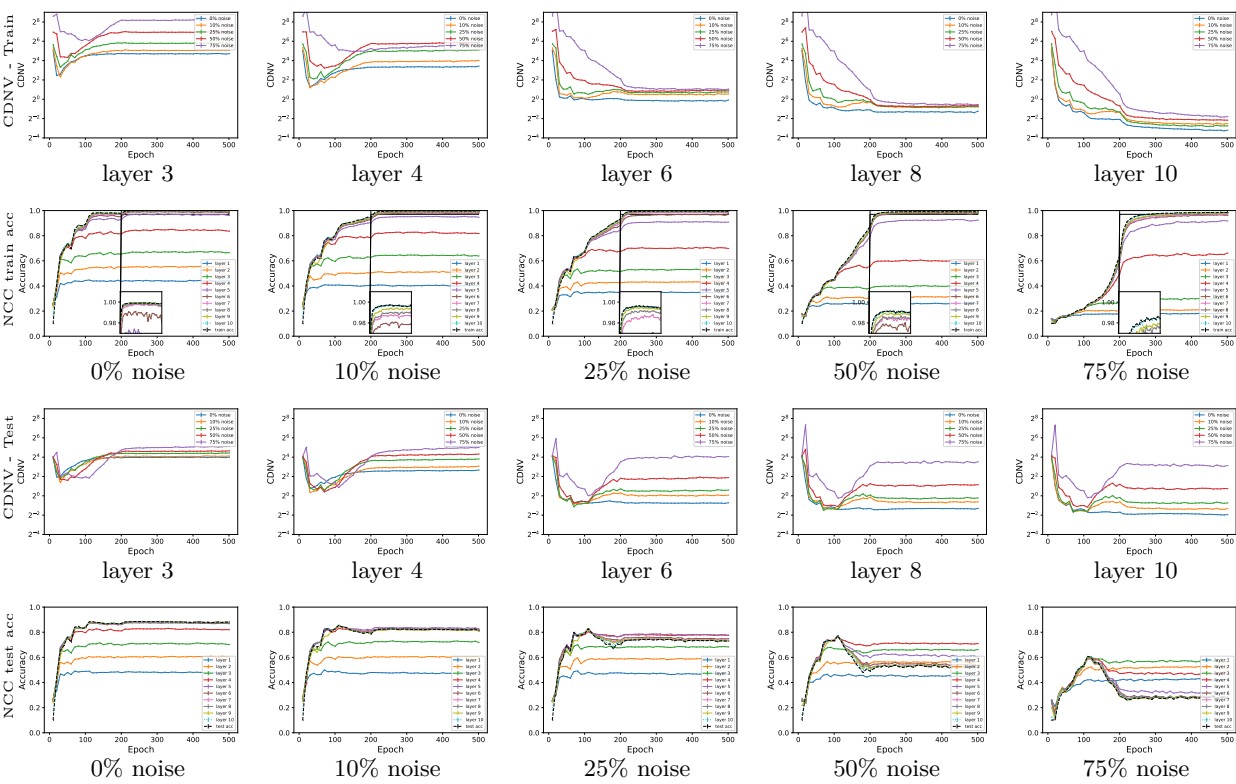

Figure 16: **Intermediate neural collapse of CONVRES-10-500 trained on CIFAR10 with noisy labels.** See Fig 3 in the main text for details.

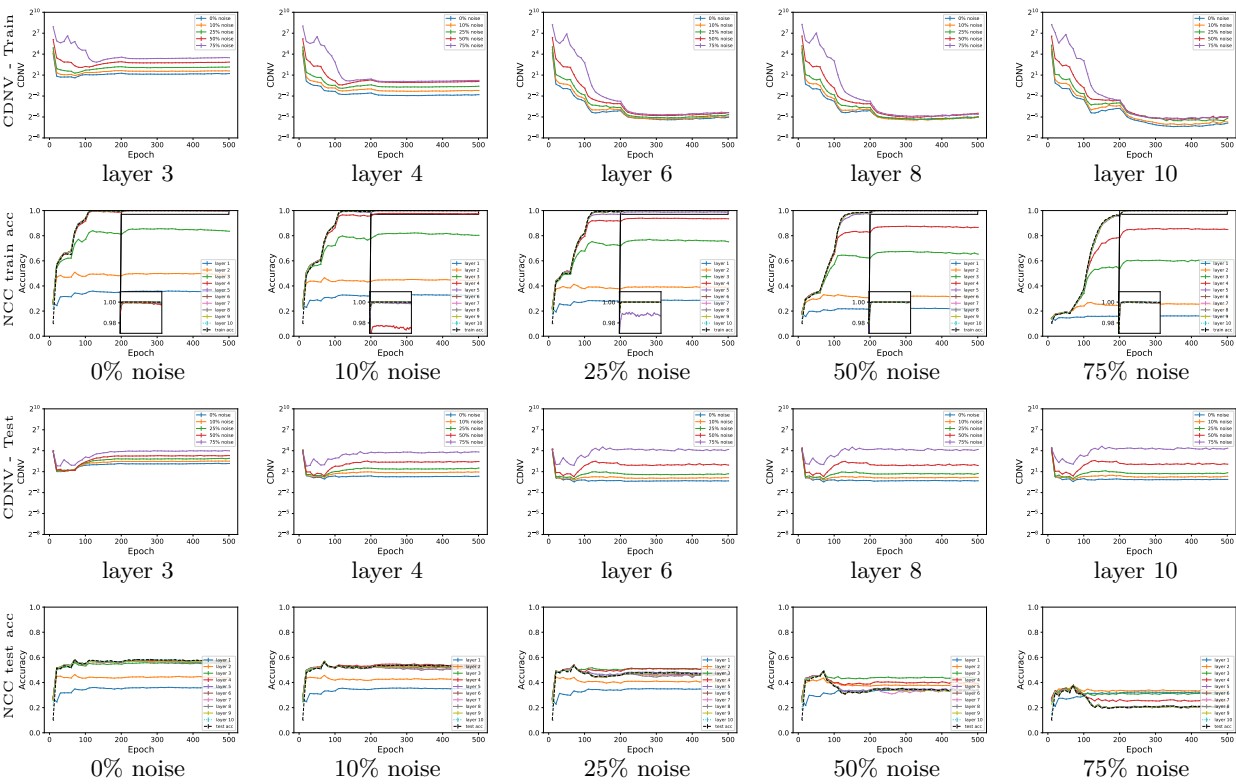

Figure 17: **Intermediate neural collapse of MLP-10-500 trained on CIFAR10 with noisy labels.** See Fig 3 in the main text for details.

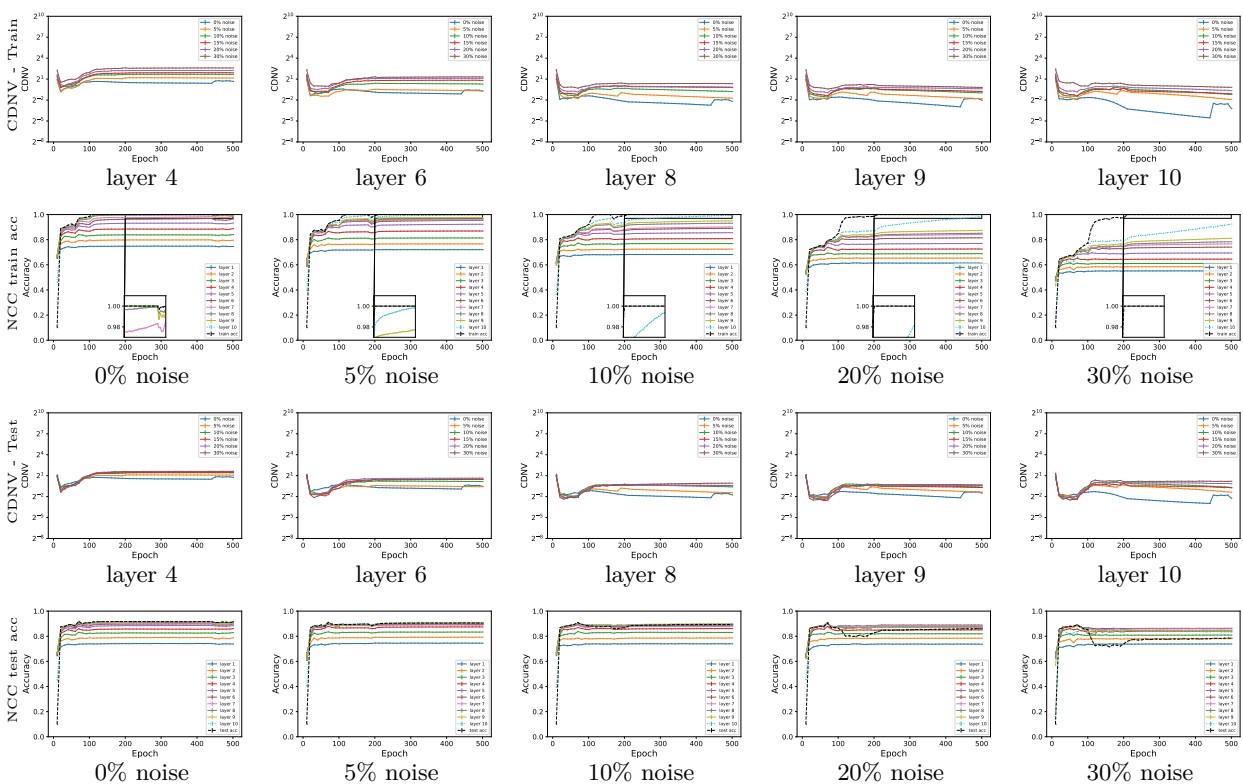

Figure 18: **Intermediate neural collapse of CONV-10-100 trained on Fashion MNIST with noisy labels.** See Fig. 3 in the main text for details.

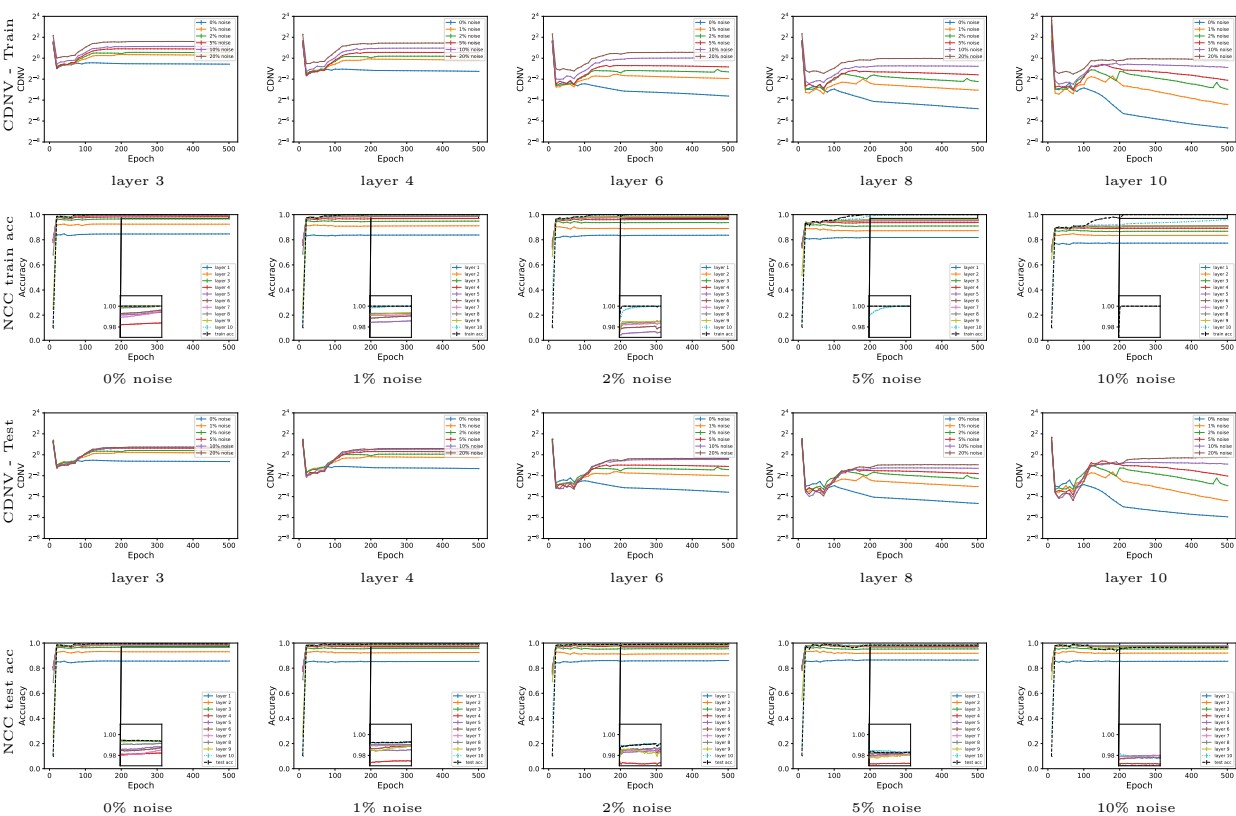

Figure 19: **Intermediate neural collapse of CONV-10-50 trained on MNIST with partially corrupted labels.** See Fig. 15 for details.

# B    Proofs

**Proposition 3.** *Let $m \in \mathbb{N}$, $p \in (0, 1/2)$, $\alpha \in (0, 1)$ and $\epsilon \in (0, 1)$. Assume that the error of the learning algorithm is $\delta_m^1$-uniform. Assume that $S_1, S_2 \sim P_B(m)$. Let $h_{S_1}^{W_0}$ be the output of the learning algorithm given access to a dataset $S_1$ and initialization $W_0$. Then,*

$$
\mathbb{E}_{S_1} \mathbb{E}_{W_0 \sim Q}[\mathrm{err}_P(h_{S_1}^{W_0})] \;\leq\; \mathbb{P}_{S_1, S_2, \tilde{Y}_2}\left[\mathbb{E}_{W_0 \sim Q}[\mathscr{d}_{S_1}^\epsilon(h_{S_1}^{W_0})] \;\geq\; \mathscr{d}_{\min}^\epsilon(\mathcal{G}, S_1 \cup \tilde{S}_2)\right]
$$
$$
+ (1 + \alpha) p + \delta_m^1 + \delta_{m,p,\alpha}^2,
\tag{6}
$$

*where $\tilde{Y}_2 = \{\tilde{y}_i\}_{i=1}^m$ denotes a set of random labels that are generated by first taking the set $Y_2$, then randomly selecting $\lfloor pm \rfloor$ labels from this set, each of the selected labels is re-selected randomly from the set $[k]$.*

*Proof.* Let $S_1 = \{(x_i^1, y_i^1)\}_{i=1}^m$ and $S_2 = \{(x_i^2, y_i^2)\}_{i=1}^m$ be two balanced datasets. Let $\epsilon > 0$, $p > 0$ and $q = (1 + \alpha) p$. Let $\tilde{Y}_2$ and $\hat{Y}_2$ be a uniformly selected set of labels that disagree with $Y_2$ on $\lfloor pm \rfloor$ and $\lfloor qm \rfloor$ randomly selected labels (resp.). We denote by $\tilde{S}_2$ and $\hat{S}_2$ the relabeling of $S_2$ with the labels in $\tilde{Y}_2$ and in $\hat{Y}_2$ (resp.). We define four different events,

$$
\begin{aligned}
A_1 &= \{(S_1, S_2, \tilde{Y}_2) \mid \exists\, q \geq (1 + \alpha)\, p : \; \mathscr{d}_{\min}^\epsilon(\mathcal{G}, S_1 \cup \tilde{S}_2) > \mathbb{E}_{\hat{Y}_2}[\mathscr{d}_{\min}^\epsilon(\mathcal{G}, S_1 \cup \hat{S}_2)]\} \\
A_2 &= \{(S_1, S_2) \mid \text{the mistakes of } h_{S_1}^{W_0} \text{ are not uniform over } S_2\} \\
A_3 &= \{(S_1, S_2, \tilde{Y}_2) \mid (S_1, S_2, \tilde{Y}_2) \notin A_1 \cup A_2 \text{ and } \mathbb{E}_{W_0 \sim Q}[\mathscr{d}_{S_1}^\epsilon(h_{S_1}^{W_0})] \;<\; \mathscr{d}_{\min}^\epsilon(\mathcal{G}, S_1 \cup \tilde{S}_2)\} \\
A_4 &= \{(S_1, S_2, \tilde{Y}_2) \mid (S_1, S_2, \tilde{Y}_2) \notin A_1 \cup A_2 \text{ and } \mathbb{E}_{W_0 \sim Q}[\mathscr{d}_{S_1}^\epsilon(h_{S_1}^{W_0})] \;\geq\; \mathscr{d}_{\min}^\epsilon(\mathcal{G}, S_1 \cup \tilde{S}_2)\} \\
B_1 &= \{(S_1, S_2, \tilde{Y}_2) \mid \mathbb{E}_{W_0 \sim Q}[\mathscr{d}_{S_1}^\epsilon(h_{S_1}^{W_0})] \;\geq\; \mathscr{d}_{\min}^\epsilon(\mathcal{G}, S_1 \cup \tilde{S}_2)\}
\end{aligned}
\tag{9}
$$

By the law of total expectation

$$
\begin{aligned}
\mathbb{E}_{S_1} \mathbb{E}_{W_0 \sim Q}[\mathrm{err}_P(h_{S_1}^{W_0})] &= \mathbb{E}_{S_1, S_2} \mathbb{E}_{W_0 \sim Q}[\mathrm{err}_{S_2}(h_{S_1}^{W_0})] \\
&= \sum_{i=1}^4 \mathbb{P}[A_i] \cdot \mathbb{E}_{S_1, S_2, \tilde{Y}_2}[\mathbb{E}_{W_0 \sim Q}[\mathrm{err}_{S_2}(h_{S_1}^{W_0})] \mid A_i] \\
&\leq \mathbb{P}[A_1] + \mathbb{P}[A_2] + \mathbb{E}_{S_1, S_2, \tilde{Y}_2}[\mathbb{E}_{W_0 \sim Q}[\mathrm{err}_{S_2}(h_{S_1}^{W_0})] \mid A_3] + \mathbb{P}[B_1],
\end{aligned}
\tag{10}
$$

where the last inequality follows from $\mathrm{err}_{S_2}(h_{S_1}^{W_0}) \leq 1$, $\mathbb{P}[A_3] \leq 1$ and $A_4 \subset B_1$.

We would like to upper bound each one of the above terms. First, we notice that since the mistakes of the network are $\delta_m^1$-uniform, $\mathbb{P}[A_2] \leq \delta_m^1$. In addition, by definition $\mathbb{P}[A_1] \leq \delta_{m,p,\alpha}^2$.

As a next step, we upper bound $\mathbb{E}_{S_1, S_2, \tilde{Y}_2}[\mathbb{E}_{W_0 \sim Q}[\mathrm{err}_{S_2}(h_{S_1}^{W_0})] \mid A_3]$. Assume that $(S_1, S_2, \tilde{Y}_2) \in A_3$. Hence, $(S_1, S_2, \tilde{Y}_2) \notin A_1 \cup A_2$. Then, the mistakes of $h_{S_1}^{W_0}$ over $S_2$ are uniformly distributed (with respect to the selection of $W_0$). Assume by contradiction that $\mathrm{err}_{S_2}(h_{S_1}^{W_0}) > (1 + \alpha)\, p$ with non-zero probability over the selection of $W_0$. Then, since the mistakes of $h_{S_1}^{W_0}$ over $S_2$ are uniformly distributed, $\mathrm{err}_{S_2}(h_{S_1}^{W_0}) > (1 + \alpha)\, p$ for all values of $W_0$. Therefore, we have

$$
\mathbb{E}_{\hat{Y}_2}[\mathscr{d}_{\min}^\epsilon(\mathcal{G}, S_1 \cup \hat{S}_2)] \;\leq\; \mathbb{E}_{W_0 \sim Q}[\mathscr{d}_{S_1}^\epsilon(h_{S_1}^{W_0})] \;<\; \mathscr{d}_{\min}^\epsilon(\mathcal{G}, S_1 \cup \tilde{S}_2),
$$

where the first inequality follows from the definition of $\mathscr{d}_{\min}^\epsilon(\mathcal{G}, S_1 \cup \hat{S}_2)$ and the second one by the assumption that $(S_1, S_2, \tilde{Y}_2) \in A_3$. However, this inequality contradicts the fact that $(S_1, S_2, \tilde{Y}_2) \notin A_1$. Therefore, we conclude that in this case, $\mathbb{E}_{W_0 \sim Q}[\mathrm{err}_{S_2}(h_{S_1}^{W_0})] \leq (1 + \alpha)\, p$ and $\mathbb{E}_{S_1, S_2, \tilde{Y}_2}[\mathbb{E}_{W_0 \sim Q}[\mathrm{err}_{S_2}(h_{S_1}^{W_0})] \mid A_3] \leq (1 + \alpha)\, p$. $\qquad\square$

**Proposition 4.** *Let $m \in \mathbb{N}$, $p \in (0, 1/2)$, $\alpha \in (0, 1)$ and $\epsilon \in (0, 1)$. Assume that the error of the learning algorithm is $\delta_m^1$-uniform. Let $S_1, S_2, S_1^i, S_2^i \sim P_B(m)$ (for $i \in [k]$). Let $\tilde{Y}_2^i = \{\tilde{y}_i\}_{i=1}^m$ be a set of labels that disagrees with $Y_2^i$ on uniformly selected $\lfloor pm \rfloor$ labels and $\tilde{S}_2^i$ is a relabeling of $S_2$ with the labels in $\tilde{Y}_2^i$. Let*

$h_{S_1}^{W_0}$ be the output of the learning algorithm given access to a dataset $S_1$ and initialization $W_0$. Then, with probability at least $1 - \delta$ over the selection of $\{(S_1^i, S_2^i, \tilde{Y}_2^i)\}_{i=1}^k$, we have

$$\mathbb{E}_{S_1}\mathbb{E}_{W_0 \sim Q}[\mathrm{err}_P(h_{S_1}^{W_0})] \ \leq \ \frac{1}{k}\sum_{i=1}^{k} \mathbb{I}\left[\mathbb{E}_{W_0 \sim Q}[\mathscr{d}_{S_1^i}^\epsilon(h_{S_1^i}^{W_0})] \ \geq \ \mathscr{d}_{\min}^\epsilon(\mathcal{G}, S_1^i \cup \tilde{S}_2^i)\right]$$

$$+ (1+\alpha)\ p + \delta_m^1 + \delta_{m,p,\alpha}^2 + \sqrt{\frac{\log(2/\delta)}{2k}}.$$

*Proof.* By Prop. 3, we have

$$\mathbb{E}_{S_1}\mathbb{E}_{W_0 \sim Q}[\mathrm{err}_P(h_{S_1}^{W_0})] \ \leq \ \mathbb{P}_{S_1,S_2,\tilde{Y}_2}\left[\mathbb{E}_{W_0 \sim Q}[\mathscr{d}_{S_1}^\epsilon(h_{S_1}^{W_0})] \ \geq \ \mathscr{d}_{\min}^\epsilon(\mathcal{G}, S_1 \cup \tilde{S}_2)\right]$$

$$+ (1+\alpha)\ p_m + \delta_m^1 + \delta_{m,p,\alpha}^2$$

We define i.i.d. random variables

$$V_i \ = \ \mathbb{I}\left[\mathbb{E}_{W_0 \sim Q}[\mathscr{d}_{S_1^i}^\epsilon(h_{S_1^i}^{W_0})] \ \geq \ \mathscr{d}_{\min}^\epsilon(\mathcal{G}, S_1^i \cup \tilde{S}_2^i)\right]. \tag{11}$$

Therefore, we can rewrite,

$$\mathbb{P}_{S_1,S_2,\tilde{Y}_2}\left[\mathbb{E}_{W_0 \sim Q}[\mathscr{d}_{S_1}^\epsilon(h_{S_1}^{W_0})] \ \geq \ \mathscr{d}_{\min}^\epsilon(\mathcal{G}, S_1 \cup \tilde{S}_2)\right] \ = \ \mathbb{E}[V_1] \tag{12}$$

By Hoeffding's inequality,

$$\mathbb{P}\left[\left|k^{-1}\sum_{i=1}^{k} V_i - \mathbb{E}[V_1]\right| \ \geq \ \epsilon\right] \ \leq \ 2\exp(-2k\epsilon^2). \tag{13}$$

By choosing $\epsilon = \sqrt{\log(1/2\delta)/2k}$, we obtain that with probability at least $1 - \delta$, we have

$$\mathbb{E}[V_1] \ \leq \ \frac{1}{k}\sum_{i=1}^{k} V_i + \sqrt{\log(1/2\delta)/2k}. \tag{14}$$

When combined with Prop. 3, we obtain the desired bound. $\qquad\qquad\square$

**Proposition 5.** *Let $d_0, m, L \geq 3$ and let $X = \{x_i\}_{i=1}^m \subset \mathbb{R}^{d_0}$ be a set of $m$ unlabeled samples. Let $\mathcal{F}_L$ be the class of ReLU neural networks $f : \mathbb{R}^{d_0} \to \mathbb{R}^{d_0}$ of depth $L$ and width $d_0$, i.e., $f(x) = \sigma(b_L + W_L\sigma(b_{L-1} + W_{L-1}\ldots\sigma(b_1 + W_1 x)))$, where $W_i \in \mathbb{R}^{d_0 \times d_0}$ and $b_i \in \mathbb{R}^{d_0}$ for all $i \in [L]$. Let $\hat{h}_f(x) = \mathrm{sign}(\|f(x) - \mu_f(\tilde{S}_{-1})\| - \|f(x) - \mu_f(\tilde{S}_1)\|)$ be the NCC classifier associated with $f$ given $\tilde{S}$, where $\tilde{S}_c$ is the subset of $\tilde{S}$ of samples labeled $c$. Let $\tilde{Y} = (\tilde{y}_1, \ldots, \tilde{y}_m)$ be a set of $m$ uniformly distributed labels in $\{\pm 1\}$ and $\tilde{S} = \{(x_i, \tilde{y}_i)\}_{i=1}^m$. Then,*

$$L \ \geq \ \frac{C'\sqrt{m}}{d_0\log(m)\sqrt{\log(d_0)\log(em)}} \cdot \left(1 - 2\mathbb{E}_{\tilde{Y}}\left[\min_{f \in \mathcal{F}_L} \mathrm{err}_{\tilde{S}}(\hat{h}_f)\right]\right) \tag{15}$$

*for some universal constant $C' > 0$.*

*Proof.* We define a set of classifiers $\mathcal{H}' = \{h(x) = \mathrm{sign}(\|f(x) - \mu_1\| - \|f(x) - \mu_2\|) \mid f \in \mathcal{F}_L, \mu_1, \mu_2 \in \mathbb{R}^{d_0}\}$. By definition, the Rademacher complexity of $\mathcal{H}'$ can be written as follows (see (Mohri et al., 2018)):

$$\mathrm{Rad}_X(\mathcal{H}') \ = \ \frac{1}{m}\mathbb{E}_{\tilde{Y}}\sup_{h \in \mathcal{H}'}\sum_{i=1}^{m} \tilde{y}_i h(x_i) \tag{16}$$

We notice that $\tilde{y}_i h(x_i) = 1$ if $h(x_i) = \tilde{y}_i$ and $\tilde{y}_i h(x_i) = -1$ otherwise. Therefore $\frac{1}{2}(\tilde{y}_i h(x_i)+1) = \mathbb{I}[h(x_i) = \tilde{y}_i]$ and we can represent $\tilde{y}_i h(x_i)$ using the classification error, $\tilde{y}_i h(x_i) = 2\mathbb{I}[h(x_i) = \tilde{y}_i] - 1 = 2(1 - \mathbb{I}[h(x_i) \neq \tilde{y}_i]) - 1$. In particular, $\frac{1}{m}\sum_{i=1}^m \tilde{y}_i h(x_i) = 2(1 - \mathrm{err}_{\tilde{S}}(h)) - 1$. Hence,

$$\mathrm{Rad}_X(\mathcal{H}') \ = \ 2\left(1 - \mathbb{E}_{\tilde{Y}}\left[\inf_{h \in \mathcal{H}'} \mathrm{err}_{\tilde{S}}(h)\right]\right) - 1 \ \geq \ 1 - 2\mathbb{E}_{\tilde{Y}}\left[\inf_{f \in \mathcal{F}_L} \mathrm{err}_{\tilde{S}}(\hat{h}_f)\right]. \tag{17}$$

where the last inequality follows from the fact that $\hat{h}_f \in \mathcal{H}'$ for any $f \in \mathcal{F}_L$.

We notice that any $h \in \mathcal{H}'$ can be represented as follows:

$$
\begin{aligned}
h(x) &= \operatorname{sign}(\|f(x) - \mu_1\| - \|f(x) - \mu_2\|) \\
&= \operatorname{sign}(\|f(x) - \mu_1\|^2 - \|f(x) - \mu_2\|^2) \\
&= \operatorname{sign}(\langle f(x), 2(\mu_2 - \mu_1)\rangle + \|\mu_1\|^2 - \|\mu_2\|^2).
\end{aligned}
$$

Therefore, $\mathcal{H}' \subset \mathcal{H} = \{h(x) = \operatorname{sign}(\langle f(x), w\rangle + b) \mid f \in \mathcal{F}_L, w \in \mathbb{R}^{d_0}, b \in \mathbb{R}\}$. In particular, $\operatorname{Rad}_S(\mathcal{H}') \leq \operatorname{Rad}_S(\mathcal{H})$. By (cf. Mohri et al. (2018), Corollary 3.19),

$$
\sqrt{\frac{2\dim(\mathcal{H})\log(em)}{m}} \geq \operatorname{Rad}_X(\mathcal{H}) \geq 1 - 2\mathbb{E}_{\tilde{Y}}\left[\inf_{f \in \mathcal{F}_L} \operatorname{err}_{\tilde{S}}(\hat{h}_f)\right]. \tag{18}
$$

Furthermore, by (Bartlett et al., 2019), $\dim(\mathcal{H}) = \mathcal{O}(UL\log(U))$, where $U$ is the total number of trainable parameters in $h \in \mathcal{H}$. By counting the number of parameters, we obtain that $U = 1 + d_0 + d_0^2 + Ld_0 + (L-1)d_0^2 = \mathcal{O}(Ld_0^2)$. Therefore, there exists some constant $\alpha > 1$, such that $\dim(\mathcal{H}) \leq \alpha(L^2 d_0^2 \log(L)^2 \log(d_0))$. Hence, we obtain that

$$
L\log(L) \geq \frac{C'\sqrt{m}}{d_0\sqrt{\log(d_0)\log(em)}} \cdot \left(1 - 2\mathbb{E}_{\tilde{Y}}\left[\inf_{f \in \mathcal{F}_L} \operatorname{err}_{\tilde{S}}(\hat{h}_f)\right]\right), \tag{19}
$$

for some universal constant $C' \in (0,1)$.

As a next step, we would like to prove that $L \geq a/\log(m)$, where $a$ denotes the right-hand side of equation 19. We consider two cases; $a \leq e$ and $a \geq e$ (where $e$ is Euler's number).

If $a \leq e$, since $L \geq 3$ and $m \geq 3$, we immediately obtain that $L \geq e \geq a/\log(m)$.

On the other hand, if $a \geq e$, we can use the following inequality. Suppose $L, a$ are two real numbers, such that, $L\log(L) \geq a \geq e$, then we have $L \geq \exp(W(a)) \geq \exp(\log(a) - \log\log(a)) \geq a/\log(a)$, where $W$ is Lambert's function. Since $a \geq e$ and $a \leq m$ (since $d_0 \geq 3$, $m \geq 3$ and $C'$, $(1 - 2\mathbb{E}_{\tilde{Y}}[\operatorname{err}_{\tilde{S}}(\hat{h}_f)]) \leq 1$), we have, $\log(a) \leq \log(m)$ and $L \geq a/\log(m)$ as desired. $\qquad\square$

**Proposition 1.** *Let $d_0, m, L \geq 3$ and let $X = \{x_i\}_{i=1}^m \subset \mathbb{R}^{d_0}$ be a set of $m$ unlabeled samples. Let $\mathcal{G} = \{\sigma(Wx + b) \mid W \in \mathbb{R}^{d_0 \times d_0}, b \in \mathbb{R}^{d_0}\}$, where $\sigma$ is the ReLU element-wise activation function. Let $\tilde{Y} = (\tilde{y}_1, \ldots, \tilde{y}_m)$ be a set of $m$ uniformly distributed labels in $\{\pm 1\}$ and $\tilde{S} = \{(x_i, \tilde{y}_i)\}_{i=1}^m$. Then,*

$$
\mathbb{E}_{\tilde{Y}}[\mathscr{d}_{\min}^\epsilon(\mathcal{G}, \tilde{S})] \geq \frac{C\sqrt{m} \cdot (1 - 2\epsilon)^2}{d_0\log(m)\sqrt{\log(d_0)\log(em)}} \tag{4}
$$

*for some universal constant $C > 0$.*

*Proof.* Let $\delta \in (0,1)$ and let $L = \left\lfloor \frac{\mathbb{E}[\mathscr{d}_{\min}^\epsilon(\mathcal{G}, \tilde{S})]}{\delta} \right\rfloor$. Since $\mathscr{d}_{\min}^\epsilon(\mathcal{G}, \tilde{S})$ is non-negative, we can apply Markov's inequality,

$$
\mathbb{P}\left[\mathscr{d}_{\min}^\epsilon(\mathcal{G}, \tilde{S}) \leq \left\lfloor \frac{\mathbb{E}[\mathscr{d}_{\min}^\epsilon(\mathcal{G}, \tilde{S})]}{\delta} \right\rfloor\right] = \mathbb{P}\left[\mathscr{d}_{\min}^\epsilon(\mathcal{G}, \tilde{S}) \leq \frac{\mathbb{E}[\mathscr{d}_{\min}^\epsilon(\mathcal{G}, \tilde{S})]}{\delta}\right] \geq 1 - \delta, \tag{20}
$$

where the probability is taken with respect to the selection of $\tilde{Y}$. The first equation follows from the fact that $\mathscr{d}_{\min}^\epsilon(\tilde{S})$ is an integer.

By Prop. 5,

$$
\frac{\mathbb{E}[\mathscr{d}_{\min}^\epsilon(\mathcal{G}, \tilde{S})]}{\delta} \geq L \geq \frac{C'\sqrt{m}}{d_0\log(m)\sqrt{\log(d_0)\log(em)}} \cdot \left(1 - 2\mathbb{E}_{\tilde{Y}}\left[\inf_{f \in \mathcal{F}_L} \operatorname{err}_{\tilde{S}}(\hat{h}_f)\right]\right). \tag{21}
$$

As a next step, we would like to bound $\mathbb{E}_{\tilde{Y}}[\inf_{f \in \mathcal{F}_L} \operatorname{err}_{\tilde{S}}(\hat{h}_f)]$. We note that any $L$-layer ReLU fully-connected network can be implemented using an $(L+1)$-layer ReLU fully-connected network (simply by taking the last layer to be a fully-connected layer with zero bias and identity weight matrix). Therefore, $\mathcal{F}_{L_1} \subset \mathcal{F}_{L_2}$ for all

$L_1 \leq L_2$. In particular, if $L^* = \ell^\epsilon_{\min}(\mathcal{G}, \tilde{S}) \leq L$, then $\inf_{f \in \mathcal{F}_L} \mathrm{err}_{\tilde{S}}(\hat{h}_f) \leq \inf_{f \in \mathcal{F}_{L^*}} \mathrm{err}_{\tilde{S}}(\hat{h}_f) \leq \epsilon$. We note that $L^* \leq L$ occurs with probability at least $1 - \delta$ by equation 20. In case this event does not happen, we may just use the fact that $\inf_{f \in \mathcal{F}_L} \mathrm{err}_{\tilde{S}}(\hat{h}_f) \leq 1$. Therefore, by the law of total expectation:

$$\mathbb{E}_{\tilde{Y}}[\inf_{f \in \mathcal{F}_L} \mathrm{err}_{\tilde{S}}(\hat{h}_f)] \leq (1 - \delta) \cdot \epsilon + \delta \cdot 1 \leq \epsilon + \delta. \tag{22}$$

Therefore, for any $\delta \in (0, 1)$, we have

$$\mathbb{E}[\ell^\epsilon_{\min}(\mathcal{G}, \tilde{S})] \geq \frac{C'\sqrt{m}}{d_0 \log(m)\sqrt{\log(d_0)\log(em)}}(1 - 2(\epsilon + \delta))\delta. \tag{23}$$

By picking $\delta = (1 - 2\epsilon)/4$, we get $(1 - 2(\epsilon + \delta))\delta = (1 - 4\epsilon + 4\epsilon^2)/8 = (1 - 2\epsilon)^2/8$ and we finally obtain that

$$\mathbb{E}[\ell^\epsilon_{\min}(\mathcal{G}, \tilde{S})] \geq \frac{C\sqrt{m} \cdot (1 - 2\epsilon)^2}{d_0 \log(m)\sqrt{\log(d_0)\log(em)}} \tag{24}$$

for $C = C'/8$. $\qquad\square$

**Proposition 2.** *Let $d_0, m, L \geq 3$, and let $S_1, S_2 \subset \mathbb{R}^d \times \{\pm 1\}$ be two sets of $m$ labeled samples. Let $\tilde{Y}_2$ be labels that are generated by first taking the set $Y_2$ (the labels of $S_2$), then randomly selecting $\lfloor pm \rfloor$ labels from this set. Each of the selected labels is then replaced with a random selection from the set $\{\pm 1\}$. Let $\tilde{S}_2$ be the same set as $S_2$ with the labels $Y_2$ replaced with $\tilde{Y}_2$. Then,*

$$\mathbb{E}_{\tilde{Y}_2}[d^0_{\min}(\mathcal{G}, S_1 \cup \tilde{S}_2)] \geq \frac{C\sqrt{\lfloor pm \rfloor}}{d_0 \log(\lfloor pm \rfloor)\sqrt{\log(d_0)\log(e\lfloor pm \rfloor)}} \tag{5}$$

*for some universal constant $C > 0$.*

*Proof.* Let $S_1, S_2$ be two sets of $m$ labeled samples. Let $X_2$ be the set of unlabeled version of $S_2$. We note that $\tilde{Y}_2$ is generated by first selecting a subset $X_3$ of $\lfloor pm \rfloor$ unlabeled samples and relabeling them with random labels $\tilde{Y}_3$ from $\{\pm 1\}$. This gives a new set $\tilde{S}_3$ of $\lfloor pm \rfloor$ randomly labeled samples.

Let $f$ be a neural network of depth $d^0_{\min}(\mathcal{G}, S_1 \cup \tilde{S}_2)$ such that the NCC classifier $\hat{h}_f$ (given $S_1 \cup \tilde{S}_2$) has NCC train error 0. Since $\hat{h}_f$ correctly classifies all of the samples in $S = S_1 \cup \tilde{S}_2$, we notice that $\hat{h}_f$ correctly classifies the subset $\tilde{S}_3$ of the $\lfloor pm \rfloor$ randomly labeled samples in $\tilde{S}_2$. Hence,

$$\mathbb{E}_{\tilde{Y}_2}[d^0_{\min}(\mathcal{G}, S_1 \cup \tilde{S}_2)] \geq \mathbb{E}_{X_3}\mathbb{E}_{\tilde{Y}_3}[d^0_{\min}(\mathcal{G}, \tilde{S}_3)]. \tag{25}$$

By Lem. 1, for any set $X_3$ of unlabeled samples, we have

$$\mathbb{E}_{\tilde{Y}_3}[d^0_{\min}(\mathcal{G}, \tilde{S}_3)] \geq \frac{C\sqrt{\lfloor pm \rfloor}}{d_0 \log(\lfloor pm \rfloor)\sqrt{\log(d_0)\log(e\lfloor pm \rfloor)}}. \tag{26}$$

Hence,

$$\mathbb{E}_{\tilde{Y}_2}[d^0_{\min}(\mathcal{G}, S_1 \cup \tilde{S}_2)] \geq \mathbb{E}_{X_3}\mathbb{E}_{\tilde{Y}_3}[d^0_{\min}(\mathcal{G}, \tilde{S}_3)] \geq \frac{C\sqrt{\lfloor pm \rfloor}}{d_0 \log(\lfloor pm \rfloor)\sqrt{\log(d_0)\log(e\lfloor pm \rfloor)}}, \tag{27}$$

as desired. $\qquad\square$

