# OpenReview forum: "Comparative Generalization Bounds for Deep Neural Networks"
_TMLR — Accepted by TMLR_

### Review · Reviewer_Pgfg · 2023-03-11

**Summary Of Contributions:**

This paper theoretically and empirically studies generalization bounds of deep neural networks (including convolutional, residual convolutional and feedforward architectures) in classification problems. The most outstanding contribution is the obtained generalization bound is much tighter than existing norm-based or uniform concentration-based generalization bounds (as demonstrated in Table 1). A closer examination of the generalization error in Equation (4) reveals several intriguing characteristics of the bound: 1) the bound depends on the learning algorithm and initialization; 2) the bound also depends on the data distribution; 3) the bound is intrinsic to the network architecture (e.g., effective depth), even though the network itself is overparameterized. All these observations are important factors to unveil the generalization properties of large neural networks. This reviewer believe the topics and findings in the paper will be of interest to the community.

**Audience:**

Yes

**Claims And Evidence:**

Yes

**Requested Changes:**

1. Neural collapse and effective depth are well motivated and recognized. However, conditions like $\delta$-uniform mistakes are far less explained. It gives the feeling that authors need these conditions because they know how to do the proof under these conditions. At least, some empirical demonstration should be provided to validate all the terms in the generalization bound, instead of just assuming they are dominated by the probability term.

2. NCC separability can be closely related to margin conditions in classification problems. When the noisy label becomes dominant, classification margin fades and makes separability difficult. Is it possible to prove a rigorous bound for some simplified classification problems, if general setting is elusive. A candidate can be linear separable data with a small fraction of label contamination.

3. (optional) Will the width of the network play a role in generalization? For example, is it possible that narrower networks often need large effective depth, yet wider networks do not (or there is a threshold on the width of the network).

**Strengths And Weaknesses:**

Apart from the contributions discussed in the previous section, the paper is generally well prepared and easy to follow. Many experimental details are discussed both in the paper and appendix. These results either support the theory or better demonstrate key concepts, e.g., effective depth, in the paper.

While the study is interesting and the theoretical results are correct, this reviewer has a mixed evaluation towards the empirical and theoretical results. Specifically, the generalization theory in proposition 1 seems like a proof-of-concept, in that the four terms in the right-hand side of Equation (4) are all elusive. It is rather difficult to assess the magnitude of these terms theoretically and rigorously; moreover, some quantities are assumed to be small without verification, e.g., $\delta_{m, p, \alpha}$ and $\delta_m^1$.

On the other hand, experimental results are much comprehensive compared to theoretical results. The test of effective depth makes sense, and the predicted generalization bounds indeed tightens all existing ones significantly. This reviewer appreciates that details of how to estimate the first probability term in the generalization bound are provided in appendix. However, it misses any test of the $\delta_m$-uniform mistakes assumption in experiments.

---

> ### Author Response · Authors · 2023-03-25
> **Response**
>
> > Reviewer: Neural collapse and effective depth are well motivated and recognized. However, conditions like $\delta$-uniform mistakes are far less explained. It gives the feeling that authors need these conditions because they know how to do the proof under these conditions. At least, some empirical demonstration should be provided to validate all the terms in the generalization bound, instead of just assuming they are dominated by the probability term.
>
> These conditions are indeed used to simplify the proofs of the main theoretical results. To further validate these assumptions, we ran several experiments that are described in Sec. 4 (see the red part in Sec. 4.2).
>
> > Reviewer: NCC separability can be closely related to margin conditions in classification problems. When the noisy label becomes dominant, classification margin fades and makes separability difficult. Is it possible to prove a rigorous bound for some simplified classification problems, if general setting is elusive. A candidate can be linear separable data with a small fraction of label contamination.
>
> We agree with the reviewer, when increasing the amount of noisy labels, classification margin and NCC separability decay. This is an observation we make in the paper (For example, Figs. 3, 14, 15) and we would like to note that this observation is in fact a positive thing in our analysis.
>
> Furthermore, in settings where NCC separability does not happen, one could easily adjust the notions of complexity (effective depth and minimal depth) by considering other notions of separability. For example, we could redefine the effective depth (Def. 1) to be the minimal layer of the network, for which the data becomes linearly separable up to a predefined margin $\Delta$ and similarly to define the minimal depth (Def. 2) as the minimal number of layers of a network that projects the data to linearly separable embeddings with margin $\Delta$.
>
> We use the notion of NCC separability for two main reasons. First, it is a cost-effective method of achieving separability as it does not involve actively training a linear separator on the data. Training a linear separator requires making certain choices, such as selecting the algorithm for training the linear classifier, its hyperparameters, and the margin, which can lead to obtaining sub-optimal linear classifiers.
> Secondly, it appears that NCC separability is present in intermediate layers of standard neural networks (such as MLPs, ConvNets, and ResNets with BatchNorm layers) that are trained using standard learning settings. This observation together with the appealing behavior of the minimality hypothesis as we observe, provide us with excellent generalization bounds even for deep networks.
>
> > Reviewer: (optional) Will the width of the network play a role in generalization? For example, is it possible that narrower networks often need large effective depth, yet wider networks do not (or there is a threshold on the width of the network).
>
> Our proposed bound relies on comparing the complexity of the network trained on correct labels to the minimal complexity required to fit partially corrupted labels. It is likely that narrower networks will have higher effective depth, but this may also hold true in the case of corrupted labels. However, due to time constraints, we have not been able to thoroughly investigate the effects of varying widths on the bound. We hope to explore this in future work.

---

### Review · Reviewer_rPZ8 · 2023-03-12

**Summary Of Contributions:**


This work investigates DNN generalization by introducing a new measure of the effective depth. It is defined as the first layer at which the sample embeddings are separable using a nearest-class classifier.

As per their bound, a network will generalize if its complexity is small compared to the complexity of a network required to fit the dataset with partially random labels.

This work provides the following key observations and validates these through empirical evaluations on multi-class classification datasets and various architectures.
- Effective depth of the trained networks increase as the amount of random labels in the data increases
- DNNs tend to converge to an effective depth $L_0$ regardless of their actual depth $L$ ( any feature layer above depth $L_0$ is nearest-class  )
- Their proposed generalization bound is empirical, non-vacuous, and independent of the network depth.


**Audience:**

Yes

**Claims And Evidence:**

Yes

**Requested Changes:**

Questions for Authors:
------------

- Table 1 does not include non-vacuous bounds provided by recent works (see missing Related works)

- In Figure 1, what are the vertical and horizontal lines at the Epoch=200 mark?

- Since you already split the data into sets S1 and S2, and train multiple networks. Could you report the performance of the cross-validation term as a baseline on the generalization error (simply the average of the error on the set S2 achieved by various networks when trained on set S1)?

- Does your bound suggest anything w.r.t. improvements on the DNN architectures (since their effective depth is more often than not much smaller than their actual depth)?

- How does your bound adapt to the setup where the later layers do not have the same width (as is prevalent in many popular architectures)?

- How do you scale the proposed generalization measure to larger datasets (I would expect training multiple hypothesis on a large scale would be computationally expensive)?

Missing Related Works:
---------

- Non-Vacuous Generalisation Bounds for Shallow Neural Networks (https://arxiv.org/pdf/2202.01627.pdf)
- Computing Nonvacuous Generalization Bounds for Deep (Stochastic) Neural Networks with Many More Parameters than Training Data (https://arxiv.org/pdf/1703.11008.pdf)
- NON-VACUOUS GENERALIZATION BOUNDS AT THE IMAGENET SCALE: A PAC-BAYESIAN COMPRESSION APPROACH (https://openreview.net/pdf?id=BJgqqsAct7)
- A New Family of Generalization Bounds Using Samplewise Evaluated CMI (https://openreview.net/pdf?id=GRd5UCkkXcV)


**Strengths And Weaknesses:**


Strengths:
-----------

- Empirical validation of the observation that DNNs tend to converge to an effective depth $L_0$ regardless of their actual depth $L$
- Ability to provide non-vacuous generaliation bounds

Weaknesses:
-----------

- Missing discussion and comparison with other non-vacuous bounds using PAC-Bayes or Conditional Mutual Information (see Missing Related works)

- Since the proposed generalization bound requires splitting the training data into two sets and training multiple networks on this split. It would also be meaningful to include the average of the cross-validation error across these runs as a baseline for the estimated generalization error.

- Estimating the generalization bound would require many computationally expensive runs (which might beg the question, why not just train the network for long enough and estimate the cross-validation error?)

---

> ### Author Response · Authors · 2023-03-25
> **Response**
>
> > Reviewer: Missing discussion and comparison with other non-vacuous bounds using PAC-Bayes or Conditional Mutual Information (see Missing Related works)
>
> We would like to thank the reviewer for drawing our attention to the papers they mentioned. We added a paragraph in Section 1 discussing these papers (highlighted in brown). While these papers are relevant, we encountered several theoretical and experimental challenges in comparing our results with theirs, which we describe below and address in Section 4.2 (highlighted in brown).
>
> Among those challenges, while our bounds apply to deep networks, Biggs & Guedj (2022) only consider 2-layer networks. Dziugaite & Roy (2017), Hellstrom & Durisi (2022) and Zhou et al. (2022) adopt a PAC-Bayesian framework that incorporates random classifiers, which is different from our approach. In addition, the experiments of Dziugaite & Roy (2017) are based on a binary version of MNIST, where digits 0-4 are classified as 0, and digits 5-9 are classified as 1. Similarly, Hellstrom & Durisi (2022) examine experiments on MNIST and CIFAR10, but only on a binary version of MNIST that includes digits 4 and 9. They also pretrain their model on ImageNet before fine-tuning it on CIFAR10, which may have influenced the test performance and the generalization bounds, particularly as CIFAR10 is a subset of ImageNet. Zhou et al. (2022) report their bound without mentioning the test performance of their model.
>
> Given these variations, it is challenging to compare our results with those of previous studies and we provided a broad comparison of best-performing settings of each bound (see Tab. 3).
>
> > Reviewer: Estimating the generalization bound would require many computationally expensive runs (which might beg the question, why not just train the network for long enough and estimate the cross-validation error?)
>
> One could estimate the performance of the output model through cross-validation or by estimating expected performance on a held-out test/validation set. In fact, one could replace any generalization bound by cross-validation or estimating the performance on a held-out set. However, while these approaches can provide generally accurate estimates of the test performance, they do not necessarily offer insights into why a model performs well or not.
>
> On the other hand, our bound provides insights on the reasons behind good generalization. Informally, it demonstrates that if the learning algorithm tends to select models of relatively small effective depths, the model will provably generalize well. Empirical evidence supports this finding, which holds true in many practical settings. Therefore, our approach provides valuable insights into understanding generalization, beyond what is provided by cross-validation or estimating expected performance on held-out test/validation sets.
>
> > Reviewer: Since the proposed generalization bound requires splitting the training data into two sets and training multiple networks on this split. It would also be meaningful to include the average of the cross-validation error across these runs as a baseline for the estimated generalization error.
>
> We agree that the cross-validation error may be a good baseline for the generalization error. However, due to time constraints, computing the cross-validation error requires running hundreds of experiments, as provided throughout the paper. The experiments are run for different initialization seeds and  different $S_1, S_2$ splits to justify the correctness of the bound, as well as the confidence in the test error. Given that both $S_2$, and the test set samples of the same distribution, we believe that the test error should be a solid baseline for the expected error. Additionally, we added the standard deviation of the test error to Table 1, to show that the test errors barely vary between different initializations.
>
> > Reviewer: In Figure 1, what are the vertical and horizontal lines at the Epoch=200 mark?
>
> The small rectangle located at the bottom of each plot provides a magnified view of the lines between epochs 200 to 500, specifically within the y-axis range of 0.97 to 1.1. A vertical line connects the small rectangle with the zoomed-in area, while a horizontal line indicates the minimum value of y, which is set at y=0.97. Following the reviews, we added a comment about it (please refer to the blue part at the introduction of Section 4).

---

> ### Author Response · Authors · 2023-03-25
> **Response contd.**
>
> > Reviewer: Does your bound suggest anything w.r.t. improvements on the DNN architectures (since their effective depth is more often than not much smaller than their actual depth)?
>
> Although the bound itself does not directly impact architecture design, it does suggest that, when designing a given architecture, we should prioritize learning algorithms and hyperparameters that minimize the effective depth of the network.
> > Reviewer: How does your bound adapt to the setup where the later layers do not have the same width (as is prevalent in many popular architectures)?
>
> Suppose we consider a model with $L$ layers and maximal width $n_0$, where the width can differ between layers, extending our bound to this case is straightforward. The effective depth of the architecture would still be measured as the first layer for which NCC separability occurs. The minimal complexity would be the minimal number of layers of width $n_0$ required to achieve NCC separability up to an error of $\epsilon$. We would then apply the bound with the original notions of effective and minimal depths replaced by the new ones. We added a comment (blue) on that at the end of Sec 3.2.
>
> > Reviewer: How do you scale the proposed generalization measure to larger datasets (I would expect training multiple hypothesis on a large scale would be computationally expensive)?
>
> We acknowledge the reviewer's concern regarding the computational expense of estimating our bound when training on large-scale datasets. Indeed, this is a limitation of our work. However, the primary objective of our study is to introduce a novel theoretical perspective on generalization and generalization bounds, by comparing the complexity of networks trained with and without noisy labels.
> Although we recognize that our experimental approach may not be readily scalable to larger datasets, it is worth noting that the majority of research on generalization bounds struggles to compute non-vacuous bounds even for relatively small datasets such as MNIST (please refer to the comments above). Our study demonstrates that our bound can provide non-vacuous bounds for the CIFAR10 dataset (see Tab. 1).

---

> > ### Comment · Reviewer_rPZ8 · 2023-04-19
> > **Rebuttal Response**
> >
> > Thank you for answering my questions and addressing some of my concerns.

---

> > > ### Author Response · Authors · 2023-04-19
> > > **Thank you for the response**
> > >
> > > We thank the reviewer for your response. We would be happy to address any further questions or concerns.

---

### Review · Reviewer_Qs9F · 2023-03-13

**Summary Of Contributions:**

This paper provides a new view to investigate the generalization capabilities of deep neural networks. It defines the effective depth and ϵ-minimal nearest-class center (NCC) depth of neural networks, and show the relationship between these two quantities is connected with generalization. In particular, it derives a generalization bound by comparing the effective depth of a network with the minimal depth required to fit the same dataset with partially corrupted labels, under two assumptions that are well formulated. This bound provides non-vacuous predictions of test performance based on the experiments.

**Audience:**

Yes

**Broader Impact Concerns:**

No impact concerns.

**Claims And Evidence:**

Yes

**Requested Changes:**

Address the weakness 1, 2 and 3.

**Strengths And Weaknesses:**

**Strengths:**

+ This paper provides a novel view (as to me) to investigate the generalization of deep neural networks. The definition of  ϵ-effective depth and ϵ-Minimal NCC depth are elegant. More important, the generalization bound derived in this paper is non-vacuous, and this is supported by the experiments.
+ This paper provides several interesting observations empirically. E.g., it empirically demonstrates that when sufficiently deep networks are trained, they converge to the same effective depth.
+ This paper is well written. It provides clear notation for clarification, and I am glad to read it.

**Weaknesses:**

1.I wonder whether the experimental results still hold if the batch normalization (BN) layers are removed in the networks.? Indeed, the analyses in this paper will not hold if there is BN in the network, because the function of BN is different during training (using the mini-batch statistics) and test (using population statistics). Based on my understanding, the analyses shown in this paper should ensure the functions during training (for empirical errors) and test (for generalization errors) should be the same.

2.Some claims should be more rigorous. E.g., “we claimed that the ϵ-effective depth is insensitive to the actual depth of the network (once it exceeds a certain threshold).”. To support this sentence, I think this paper should consider the optimization problem that whether the deeper model can be well optimized. E.g., If this paper considering training a more complex  dataset, e.g, ImageNet, it might be another story (is it still insensitive to the actual depth of the network?); or train the 56-layer/110-layer conv network (with and without residual connection), whether the results still hold?

3.It is nice to see the proposed bound seems to be non-vacuous. However, I indeed have concerns on the effectiveness of the two assumptions. This paper should provide more discussion for the assumptions, e.g., is there any theoretically base, or can the assumption can be well investigated by experiments? It is more interesting for me, if this paper can provide experiments on synthetic datasets that can well be used to characterize the assumptions, and the results.

4.I have some concerns on the setup of initialization. It is not a common setup in practice that initializing weight $W_0=\gamma$ (a fixed number) as described in page 3 that “we initialize the weights $W_0=\gamma$ of h with a standard initialization procedure and at each iteration”. Does this paper means that $gamma$ is the variance/stdev?

---

> ### Author Response · Authors · 2023-03-25
> **Response**
>
> > Reviewer: I wonder whether the experimental results still hold if the batch normalization (BN) layers are removed in the networks.? Indeed, the analyses in this paper will not hold if there is BN in the network, because the function of BN is different during training (using the mini-batch statistics) and test (using population statistics). Based on my understanding, the analyses shown in this paper should ensure the functions during training (for empirical errors) and test (for generalization errors) should be the same.
>
> Our theoretical setting can be applied to training models with Batch Normalization layers. To make it clearer, we addressed this point after Def. 2 (see the red part).
>
> Our framework views the learning algorithm as a mapping from a dataset $S$ and initialization $W_0$ to a model $h^{W_0}_S$ that is used at test time. The bound evaluates the performance of the model $h^{W_0}_S$ (left-hand-side of Eq. 4) using a function of its effective depth (right-hand-side of Eq. 4). Therefore, our bound is agnostic to how the function $h^{W_0}_S$ was selected.
> When training a model with BN, the model uses mini-batch statistics during training. However, at the end of the training process, the algorithm returns a model $h^{W_0}_S$ that uses the population statistics. This model will be evaluated with our bound and the bound is independent of how the function $h^{W_0}_S$ was selected (whether we used mini-batch statistics during training or not). In our experiments, we followed the same process.
>
> > Reviewer: Some claims should be more rigorous. E.g., “we claimed that the $\epsilon$-effective depth is insensitive to the actual depth of the network (once it exceeds a certain threshold).”. To support this sentence, I think this paper should consider the optimization problem that whether the deeper model can be well optimized. E.g., If this paper considering training a more complex dataset, e.g, ImageNet, it might be another story (is it still insensitive to the actual depth of the network?); or train the 56-layer/110-layer conv network (with and without residual connection), whether the results still hold?
>
> The statement, "we claimed that the $\epsilon$-effective depth is insensitive to the actual depth of the network (once it exceeds a certain threshold)," informally restates the minimality hypothesis (Observation 1). Our argument is that if we can train two deep neural networks adequately, their effective depths are almost identical. However, we acknowledge that this may not be feasible if the optimization process collapses or if the networks are not optimally trained. Therefore, we have updated our statements to include this disclaimer (see the red part after Def. 1 and the red part at the beginning of the paragraph “The effect of the depth on the $\epsilon$-effective depth”).
> We have conducted experiments in Figure 2 to verify this claim, which demonstrate that deep networks of varying depths collapse to the same effective depth. However, it is important to note that this is ultimately a theoretical work, and our experiments serve as empirical evidence for our claims. Unfortunately, due to resource limitations, conducting large-scale experiments on ImageNet with extremely deep networks is not feasible, though such experiments would be very interesting.
>
> > Reviewer: It is nice to see the proposed bound seems to be non-vacuous. However, I indeed have concerns on the effectiveness of the two assumptions. This paper should provide more discussion for the assumptions, e.g., is there any theoretically base, or can the assumption can be well investigated by experiments? It is more interesting for me, if this paper can provide experiments on synthetic datasets that can well be used to characterize the assumptions, and the results.
>
> These conditions were used to simplify the proofs of the main theoretical results. To further validate these assumptions, we ran several experiments that are described in Sec. 4 (see the red part in Sec. 4.2).
>
> > Reviewer: I have some concerns on the setup of initialization. It is not a common setup in practice that initializing weight $W_0=\gamma$ (a fixed number) as described in page 3 that “we initialize the weights $W_0=\gamma$ of h with a standard initialization procedure and at each iteration”. Does this paper means that $\gamma$ is the variance/stdev?
>
> The variable $W_0=\gamma$ was used to represent a vector of weights selected through a standard initialization process. We understand that the use of $\gamma$ was unnecessary and created confusion. To clarify, we updated the paper and now use $W_0$ to denote the weight vector at initialization. This vector is sampled from a distribution $Q$, which could be (for example) a normal distribution or the Kaiming He initialization.

---

### Decision · Action_Editors · 2023-04-27

**Recommendation:** Accept with minor revision

**Comment:**

This paper proposes a new measure to investigate the generalization ability of neural networks. It provides a new perspective to understand the generalization by introducing effective depths and nearest-class center (NCC) depth.

In the review phase, this manuscript receives two positive and negative comments.  All the reviewers consider the proposed method novel and the experimental results comprehensive. Compared with the existing methods, this paper gives a different view and makes a contribution to the community. However, the theoretical results still have some blemishes as pointed out by reviewers (e.g., Quite a few quantities are assumed to be small but not proved to be or tested in experiments). The authors are encouraged to further improve the theoretical results in the final version.

Overall, I recommend accepting this paper with minor revision.

**Audience:**

Yes

**Claims And Evidence:**

Yes. The authors also conduct extensive experiments to investigate its effectiveness.

---

> ### Author Response · Authors · 2023-05-25
> **Camera-ready Submission**
>
> We are pleased to inform you that we have uploaded the camera-ready revision. We have incorporated the requested changes (please refer to the summary of changes), and further fixed minor typos in the manuscript. Please let us know if you have any questions or concerns. We thank the reviewers and the area chair for their continual support and constructive feedback in the reviewing process of the manuscript!